# SELF-NORMALIZED RESETS FOR PLASTICITY IN CONTINUAL LEARNING

**Vivek F. Farias**
Sloan School of Management
Massachusetts Institute of Technology
Cambridge, MA 02139, USA
`vivekf@mit.edu`

**Adam D. Jozefiak**[*]
Operations Research Center
Massachusetts Institute of Technology
Cambridge, MA 02139, USA
`jozefiak@mit.edu`

## ABSTRACT

Plasticity Loss is an increasingly important phenomenon that refers to the empirical observation that as a neural network is continually trained on a sequence of changing tasks, its ability to adapt to a new task diminishes over time. We introduce Self-Normalized Resets (SNR), a simple adaptive algorithm that mitigates plasticity loss by resetting a neuron's weights when evidence suggests its firing rate has effectively dropped to zero. Across a battery of continual learning problems and network architectures, we demonstrate that SNR consistently attains superior performance compared to its competitor algorithms. We also demonstrate that SNR is robust to its sole hyperparameter, its rejection percentile threshold, while competitor algorithms show significant sensitivity. SNR's threshold-based reset mechanism is motivated by a simple hypothesis test that we derive. Seen through the lens of this hypothesis test, competing reset proposals yield suboptimal error rates in correctly detecting inactive neurons, potentially explaining our experimental observations. We also conduct a theoretical investigation of the optimization landscape for the problem of learning a single ReLU. We show that even when initialized adversarially, an idealized version of SNR learns the target ReLU, while regularization based approaches can fail to learn.

## 1 INTRODUCTION

*Plasticity Loss* is an increasingly important phenomenon studied broadly under the rubric of continual learning (Dohare et al., 2024). This phenomenon refers to the empirical observation that as a network is continually trained on a sequence of changing tasks, its ability to adapt to a new task diminishes over time. While this is distinct from the problem of catastrophic forgetting (also studied under the rubric of continual learning (Goodfellow et al., 2013; Kirkpatrick et al., 2017)), it is of significant practical importance. In the context of pre-training language models, an approach that continually trains models with newly collected data is preferable to training from scratch (Ibrahim et al., 2024; Wu et al., 2024). On the other hand, the plasticity loss phenomenon demonstrates that such an approach will likely lead to models that are increasingly unable to adapt to new data. Similarly, in the context of reinforcement learning using algorithms like TD, where the learning tasks are inherently non-stationary, the plasticity loss phenomenon results in actor or critic networks that are increasingly unable to adapt to new data (Lyle et al., 2022). Figure 1 illustrates plasticity loss in the 'Permuted MNIST' problem introduced by Goodfellow et al. (2013).

One formal definition of plasticity measures the ability of a network initialized at a specific set of parameters to fit a random target function using some pre-specified optimization procedure. In this sense, random parameter initializations (eg. Lyle et al. (2024)) are known to enjoy high plasticity. This has motivated two related classes of algorithms that attempt to mitigate plasticity loss. The first explicitly 'resets' neurons that are deemed to have low 'utility' (Dohare et al., 2023; Sokar et al., 2023). A reset re-initializes the neurons input weights and bias according to some suitable random initialization rule, and sets the output weights to zero; algorithms vary in how the utility of a neuron is defined and estimated from online data. A second class of algorithms perform this reset procedure

---

[*]Corresponding author. Code: `https://github.com/ajozefiak/SelfNormalizedResets`.

implicitly via regularization (Ash & Adams, 2020; Kumar et al., 2023b). These latter algorithms differ in their choice of what to regularize towards, with choices including the original network initialization; a new randomly drawn initialization; or even zero. The aforementioned approaches to mitigating plasticity loss attempt to adjust the training process; other research has studied the role of architectural and optimizer hyperparameter choices. Across all of the approaches to mitigating plasticity loss described above, no single approach is yet to emerge as both robust to hyperparameter choices, and simultaneously performant across benchmark problems.

Given some point process consider the task of distinguishing between the hypotheses that this point process has a positive rate (the null hypothesis), or a rate that is identically zero with a penalty for late rejection or acceptance. An optimal test here takes the following simple form: we reject the null hypothesis as soon as the time elapsed without an event exceeds some percentile of the inter-arrival time under the null hypothesis and otherwise accept immediately upon an event. Viewing the firing of a neuron as such a point process, we propose to reset a neuron based on a rejection of the hypothesis that the the neuron is firing at a positive rate. We use the histogram of past inter-firing times as a proxy of the inter-arrival time distribution under the null hypothesis. This exceedingly simple algorithm is specified by a single hyperparameter: the rejection percentile threshold. We refer to this procedure as self-normalized resets (SNR) and argue this is a promising approach to mitigating plasticity loss:

1. We demonstrate superior performance on four benchmark problems classes studied in (Dohare et al., 2023; Kumar et al., 2023b). Interestingly, there is no single closest competitor to SNR across these problems. Many competing approaches also show significant sensitivity to the choice of hyperparameters; SNR does not. We introduce a new problem to elucidate similar plasticity loss phenomena in the context of language models, and show similar relative merits for SNR.

2. We conduct a theoretical investigation of the optimization landscape for the problem of learning a single ReLU. We show that while (an idealized version of) SNR learns the target ReLU, regularization based approaches can fail to learn in this simple setting.

## 1.1 RELATED LITERATURE

The phenomenon of plasticity loss was discovered in the context of transfer learning (Ash & Adams, 2020; Zilly et al., 2021; Achille et al., 2017). Achille et al. (2017) showed that pre-training a network on blurred CIFAR images reduces its ability to learn on the original images. In a similar vein, Ash & Adams (2020) showed that pre-training a network on 50% of a training set followed by training on the complete training set reduces accuracy relative to a network that forgoes the pre-training step. More recent literature has focused on problems that induce plasticity loss while training on a sequence of hundreds of changing tasks, such as Permuted MNIST and Continual ImageNet in Dohare et al. (2021), capturing the necessity to learn indefinitely.

**Correlates of Plasticity Loss.** The persistence of plasticity loss across a swathe of benchmark problems has elucidated a search for its cause. Several correlates of plasticity loss have been well observed, namely neuron inactivity, feature or weight rank collapse, increasing weight norms, and loss of curvature in the loss surface (Dohare et al., 2021; Lyle et al., 2023; Sokar et al., 2023; Lewandowski et al., 2023; Kumar et al., 2020). The exact cause of plasticity loss remains unclear and Lyle et al. (2023) have shown that for any correlate an experiment can be constructed in which its correlation with plasticity loss is negative. Nonetheless, these correlates have inspired a series of algorithms and interventions with varying degrees of success in alleviating the problem. However, none is consistently performant across architectures and benchmark problems.

**Reset Methods.** Algorithms that periodically reset inactive or low-utility neurons have emerged as a promising approach (Dohare et al., 2023; Sokar et al., 2023; Nikishin et al., 2022). Continual Backprop (CBP) (Dohare et al., 2023) is one such method which tracks a utility for each neuron, and according to some reset frequency $r$, it resets the neuron with minimum utility in each layer. CBP's utility is a discounted average product of a neuron's associated weights and activation, a heuristic inspired by the literature on network pruning. Another algorithm is ReDO (Sokar et al., 2023), where on every $1/r^{\text{th}}$ mini-batch, ReDO computes the average activity of each neuron and resets those neurons whose average activities are small relative to other neurons in the corresponding layer, according to a threshold hyperparameter. Two defining characteristics of CBP and ReDO are

a fixed reset rate and that neurons are reset relative to the utility of other neurons in their layer. As we will see, these proposals result in sub-optimal error rates, in a sense we make precise later.

**Regularization Methods.** L2 regularization has been shown to reduce plasticity loss, but is insufficient in completely alleviating the phenomenon (Dohare et al., 2021; Lyle et al., 2023). While L2 regularization limits weight norm growth during continual learning, it can exacerbate weight rank collapse due to regularization towards the origin. One successful regularization technique is Shrink and Perturb (S&P) (Ash & Adams, 2020), which periodically scales the network's weights by a shrinkage factor $p$ followed by adding random noise to each weight with scale $\sigma$. Another approach is to perform L2 regularization towards the initial weights referred to as L2 Init (Kumar et al., 2023b). These methods can be viewed as variants of L2 regularization that regularize towards a random initialization and the original initialization, respectively. These methods limit the growth of weight norms while maintaining weight rank and neuron activity by regularizing towards a high-plasticity parameterization.

**Architectural and Optimizer Modifications.** Architectural modifications such as layer normalization (Ba et al., 2016) and the use of concatenated ReLU activations have been shown to improve plasticity to varying degrees across network architectures and problem settings (Lyle et al., 2023; Kumar et al., 2023b). Additionally, tuning Adam hyperparameters to improve the rate at which second moment estimates are updated has been explored with some success in Lyle et al. (2023).

## 2  ALGORITHM

To make ideas precise, consider a sequence of training examples $(X_t, Y_t) \in \mathcal{X} \times \mathcal{Y}$, drawn from some distribution $\mu_t$. Denote the network by $f : \mathcal{X} \times \Theta \to \mathcal{Y}$, and let $l : \mathcal{Y} \times \mathcal{Y} \to \mathbb{R}$ be our loss function. Denote by $H_t \in \mathcal{H}_t$, the history of network weights and training examples up to time $t$, and assume access to an optimization oracle $O_t : \mathcal{H}_{t-1} \to \Theta$ that maps the history of weights and training examples to a new set of network weights. As a concrete example, $O_t$ might correspond to stochastic gradient descent.

Let $\theta_t^*$ minimize $\mathbb{E}_{\mu_t}[l(f(X_t; \theta), Y_t)]$, denote $\Theta_t = O_t(H_{t-1})$, and consider average expected regret

$$\frac{1}{T} \sum_t \mathbb{E}_{\mu_t}[l(f(X_t; \Theta_t), Y_t)] - \mathbb{E}_{\mu_t}[l(f(X_t; \theta_t^*), Y_t)]$$

*Plasticity loss* describes the phenomenon where, for certain continual learning processes $\Theta_t$, such as those corresponding to SGD or Adam, average expected regret increases over time, even for benign choices of $\mu_t$.[1] To make these ideas concrete, it is worth considering an example of the above phenomenon reported first by Dohare et al. (2021).

**Example 2.1** (The Permuted MNIST problem). Consider a sequence of 'tasks' presented sequentially to SGD, wherein each task consists of 10000 images from the MNIST dataset with the pixels permuted. SGD trains over a single epoch on each task before the subsequent task is presented. Figure 1 measures average accuracy on each task; we see that average accuracy decreases over tasks. The figure also shows a potential correlate of this phenomenon: the number of 'dead' or inactive neurons[2] in the network increases as training proceeds, diminishing the network's effective capacity.

One hypothesis that seeks to explain plasticity loss is that the network weights obtained from minimizing loss over some task yield poor initializations for a subsequent task, leading to the inactive neurons we observe in the above experiment. On the other hand random weight initializations are known to work well (Glorot & Bengio, 2010), suggesting a natural class of heuristics: re-initialize inactive neurons. Of course, the crux of any such algorithm is determining whether a neuron is inactive in the first place, and doing so as quickly as possible.

To motivate our algorithm, SNR, consider applying the network $f(\cdot; \theta_t^*)$ to a hypothetical sequence of training examples drawn i.i.d. from $\mu_t$ indexed by $s$. Let $Z_{s,i}^{\mu_t}$ indicate the sequence of activa-

---

[1]Specifically, one canonical choice of the sequence of measures $\mu_t$ considered in all of the literature on this topic is dividing $T$ into intervals, each of length, say $\Delta$, and having $\mu_t$ be constant and equal to $\mu_i$ over the $i$th such interval. If $\mu_i$ is itself drawn randomly from some distribution of measures and $\Delta$ scales faster than a constant with $T$, we would expect average expected regret to scale like a constant; this is certainly the case if the optimization problem defining $\theta_t^*$ is convex; in which case that constant is zero.

[2]this notion is formalized in Section 2.1

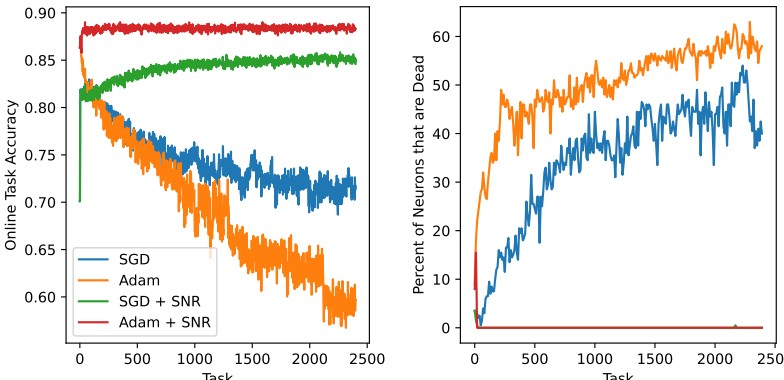

Figure 1: Illustration of plasticity loss and its mitigation by SNR during training of a multilayer perceptron on the Permuted MNIST problem for a single random seed. For this figure, a neuron is declared dead if it has not fired for the last 1000 consecutive training examples.

---

**Algorithm 1:** SNR: Self-Normalized Resets

**Input:** Reset percentile threshold $\eta$
**Initialize:** Initialize weights $\theta_0$ randomly. Set inter-firing time $a_i = 0$ for each neuron $i$
**for** each training example $x_t$ **do**
  **Forward Pass**: Evaluate $f(x_t; \theta_t)$. Get neuron activations $z_{t,i}$ for each neuron $i$
  **Update inter-firing times:** For each neuron $i$, $a_i \leftarrow a_i + 1$ if $z_{t,i} = 0$. Otherwise, $a_i \leftarrow 0$
  **Optimize**: $\theta_{t+1} \leftarrow O_t(H_t)$
  **Resets:** For each neuron $i$, reset if $\mathbb{P}(A_i^{\mu_t} \geq a_i) \leq \eta$.
**end**

---

tions of neuron $i$, and let $A_i^{\mu_t}$ be a random variable distributed as the random time between any two consecutive activations over this hypothetical sequence of examples. Now turning to the *actual* sequence of training examples, let $a_i^t$ count the time since the last firing of neuron $i$ prior to time $t$. Our (idealized) proposal is then exceedingly simple: reset neuron $i$ at time $t$ iff $\mathbb{P}(A_i^{\mu_t} \geq a_i^t) \leq \eta$ for some suitably small threshold $\eta$. We dub this algorithm *Self-Normalized Resets* and present it as Algorithm 1. The algorithm requires a single hyper parameter, $\eta$. Of course in practice, the distribution of $A_i^{\mu_t}$ is unknown to us, and so an implementable version of Algorithm 1 simply approximates this distribution with the histogram of inter-firing times of neuron $i$ prior to time $t$.[3]

## 2.1 MOTIVATING SNR AND COMPARISON TO OTHER RESET SCHEMES

Here we motivate the SNR heuristic and compare it to other proposed reset schemes. Consider the following simple hypothesis test: we observe a discrete time process $Z_s \in \{0, 1\}$ which under the null hypothesis $H_0$ is a Bernoulli process with mean $p > 0$. The alternative hypothesis $H_1$ is that the mean of the process is identically zero. A hypothesis test must, at some stopping time $\tau$, either reject ($X_\tau = 1$) or accept ($X_\tau = 0$) the null; an optimal such test would choose to minimize the sum of type-1 and type-2 errors (the 'error rate') and a penalty for delays:

$$\mathbb{P}(X_\tau = 1 | H_0) + \mathbb{P}(X_\tau = 0 | H_1) + \lambda \left( \mathbb{E}[\tau | H_0] + \mathbb{E}[\tau | H_1] \right)$$

Here the multiplier $\lambda > 0$ penalizes the delay in a decision. If $\lambda < p/2$, the optimal test takes a simple form: for some suitable threshold $\bar{T}$, reject the null iff $Z_s = 0$ for all times $s$ up to $\bar{T}$:

**Proposition 2.1.** Let $\bar{T}$ be the $1 - \lambda(p - \lambda)^{-1}$ percentile of a Geometric($p$) distribution. Then the optimal hypothesis test takes the form $X_\tau = \mathbf{1}\{Z_\tau = 0\}$ where $\tau = \min(s : Z_s = 1) \wedge \bar{T}$.

---

[3]As opposed to tracking the histogram itself, we simply track the mean inter-firing time, and assume $A_i^{\mu_t}$ is geometrically distributed with that mean. This requires tracking just one parameter per neuron. Our mean estimate is itself computed over a fixed length trailing window motivated by the change-point detection approach of Besbes & Zeevi (2011); the length of this window is a hyper-parameter.

Notice that if $\lambda \propto p$, the percentile threshold above is independent of $p$. Applying this setup to the setting where under the null, we observe the firing of neuron $i$ under i.i.d. training examples from $\mu_t$ and a neuron is considered 'dead' or inactive if the alternate hypothesis is true, imagine that $p = \mathbb{P}(Z_{s,i}^{\mu_t} = 1)$. Further, we assume $\lambda = \alpha p$ ($\alpha < 1/2$); a reasonable assumption which models a larger penalty for late detection of neurons that are highly active. It is then optimal to declare neuron $i$ 'inactive' if the length of time it has not fired exceed the $1 - \alpha(1 - \alpha)^{-1}$ percentile of the distribution of $A_i^{\mu_t}$. This is the underlying motivation for the SNR heuristic.

**Comparison with Reset Schemes:** Neuron reset heuristics such as Sokar et al. (2023) define (sometimes complex) notions of neuron 'utility' to determine whether or not to re-initialize a neuron. The utility of every neuron is computed over every consecutive (say) $r$ minibatches, and neurons with utility below a threshold are reset. To facilitate a comparison, consider the setting where neurons that do not fire at all over the course of the $r$ mini batches are estimated to have zero utility, and that only neurons with zero utility are re-initialized.

This reveals an interesting comparison with SNR. The schemes above will re-initialize a neuron after inactivity over a period of time that is *uniform* across all neurons. On the other hand, SNR will reset a neuron after it is inactive for a period that corresponds to a fixed percentile of the inter-firing time distribution of that neuron. Whereas this percentile is fixed across neurons, the corresponding period of inactivity after which a neuron is reset will vary across neurons: shorter for neurons that tend to fire frequently, and longer for neurons that fire less frequently.

We can make this comparison precise in the context of the hypothesis testing setup above: specifically, consider two neurons with null firing rates $p_1$ and $p_2$ respectively ($p_1 < p_2$), and delay multipliers, $\lambda$, of $\alpha p_1$ and $\alpha p_2$ respectively. By Proposition 2.1, under SNR, the first is reset if it is inactive for time at least $\log(\alpha(1-\alpha)^{-1})/\log(1-p_1)$ and the second if it is inactive for time at least $\log(\alpha(1-\alpha)^{-1})/\log(1-p_2)$. In contrast, for a fixed threshold scheme such as Sokar et al. (2023), either neuron would be reset after being inactive for some fixed threshold, say $r^*$. Assume $r^*$ is set to minimize the sum of the error rates of the two neurons while keeping the total delay identical to that for SNR. The proposition below compares the error rate between the two schemes:

**Proposition 2.2.** The ratio of total error rate with a fixed threshold $r^*$ to that under SNR scales like

$$\Omega\left(\exp\left(\log(\alpha(1-\alpha)^{-1})\left(-\frac{1}{2} + \frac{1}{2}\frac{\log(1-p_1)}{\log(1-p_2)}\right)\right)\right)$$

Now recall that $\alpha < 1/2$ and $p_1 < p_2$. The result above then shows that: (a) the error rate under an (optimal) fixed threshold can grow arbitrarily larger than the error rate under SNR as the penalty for delay shrinks to zero and (b) the rate at which this gap grows itself scales with the difference in the nominal firing rates of the neurons under consideration. This provides insight into the relative merits of using a scheme like SNR in lieu of existing reset proposals: *resets under SNR detect changes in the firing rate of a neuron faster and more accurately; this matters particularly in situations where there is wide disparity in the nominal firing rates of neurons across the network.*

## 3 EXPERIMENTS

We evaluate the efficacy and robustness of SNR on a series of benchmark problems from the continual learning literature, measuring regret with respect to prediction accuracy $l(y, y') = \mathbf{1}\{y \neq y'\}$. As an overview, we will seek to make the following points:

**Inactive neurons are an important correlate of plasticity loss:** This is true across several architectures: vanilla MLPs, CNNs and transformers.
**Lower average loss:** Across a broad set of problems/ architectures from the literature, SNR consistently achieves lower average loss than competing algorithms.
**No consistent second-best competitor:** Among competing algorithms, none emerge as consistently second best to SNR.
**Robustness to hyper-parameters:** The performance of SNR is robust to the choice of its single hyper parameter (the rejection percentile threshold). This is less so for competing algorithms.

## 3.1 EXPERIMENTAL SETUP

Each problem consists of tasks $\mathcal{T}_1, \mathcal{T}_2, \ldots, \mathcal{T}_N$, each of which contains training examples in $\mathcal{X} \times \mathcal{Y}$. A network is trained for a fixed number of epochs per task to minimize cross-entropy loss. We perform an initial hyperparameter sweep over 5 seeds to determine the optimal choice of hyperparameters (see Appendix C). For each algorithm and problem, we select the hyperparameters that attain the lowest average loss and repeat the experiment on 5 new random seeds. A random seed determines the network's parameter initialization, the generation of tasks, and any randomness in the algorithms evaluated. We evaluate both SGD and Adam as the base optimization algorithm, as earlier literature has argued that Adam can be less performant than SGD in some continual learning settings (Dohare et al., 2023; Ashley et al., 2021). We evaluate on the following problems:

**Permuted MNIST (PM) (Goodfellow et al., 2013; Dohare et al., 2021; Kumar et al., 2023b):** A subset of 10000 image-label pairs from the MNIST dataset are sampled for an experiment. A task consists of a random permutation applied to each of the 10000 images. The network is presented with 2400 tasks appearing in consecutive order. Each task consists of a single epoch and the network receives data in batches of size 16.
**Random Label MNIST (RM) (Kumar et al., 2023b; Lyle et al., 2023):** A subset of 1200 images from the MNIST dataset are sampled for an experiment. An experiment consists of 100 tasks, where each tasks is a random assignment of labels, consisting of 10 classes, to the 1200 images. A network is trained for 400 epochs on each task with a batch size 16.
**Random Label CIFAR (RC) (Kumar et al., 2023b; Lyle et al., 2023):** A subset of 128 images from the CIFAR-10 dataset are sampled for an experiment. An experiment consists of 50 tasks, where each tasks is a random assignment of labels, consisting of 10 classes, to the 128 images. An agent is trained for 400 epochs on each task with a batch size 16.
**Continual Imagenet (CI) (Dohare et al., 2023; Kumar et al., 2023b):** An experiment consists of all 1000 classes of images from the ImageNet-32 dataset (Chrabaszcz et al., 2017) containing 600 images from each class. Each task is a binary classification problem between two of the 1000 classes, selected at random. The experiment consists of 500 tasks and each class occurs in exactly one task. Each task consists of 1200 images, 600 from each class, and the network is trained for 10 epochs with a batch size of 100.
**Permuted Shakespeare (PS):** We propose this problem to facilitate studying the transformer architecture in analogy to the MNIST experiments. An experiment consists of 32768 tokens of text from Shakespeare's Tempest. For any task, we take a random permutation of the vocabulary of the Tempest and apply it to the text. The network is presented with 500 tasks. Each task consists of 100 epochs and the network receives data in batches of size 8 with a context widow of width 128. We evaluate over 9 seeds.

This experimental setup, for all but Permuted Shakespeare, follows that of (Kumar et al., 2023b), with the exceptions of Permuted MNIST which has its task count increased from 500 to 2400, Random Label MNIST which has its task count increased from 50 to 100, and Random Label CIFAR which has its dataset reduced from 1200 to 128 images. Lyle et al. (2023) consider variants of the Random Label MNIST and CIFAR problems by framing them as MDP environments for DQN agents. During training, the DQN agents are periodically paused to assess their plasticity by training them on separate, randomly generated regression tasks using the same image datasets.

### 3.1.1 ALGORITHMS AND ARCHITETCURES

Our baseline in all problems consist simply of using SGD or Adam as the optimizer with no further intervention. We then consider several interventions to mitigate plasticity loss. First, we consider algorithms that employ an explicit reset of neurons: these include SNR, Continual Backprop (CBP) (Dohare et al., 2021), and ReDO (Sokar et al., 2023). Among algorithms that attempt to use regularization, we consider vanilla L2 regularization, L2 Init (Kumar et al., 2023b), and Shrink and Perturb (Ash & Adams, 2020). Finally, as a potential architectural modification we consider the use of Layer Normalization (Ba et al., 2016).

We utilize the following network architectures:
**MLP**: For Permuted MNIST and Random Label MNIST we use an MLP identical to that in Kumar et al. (2023b) which in turn is a slight modification to that in Dohare et al. (2023).
**CNN**: For Random Label CIFAR and Continual ImageNet we use a CNN architectures identical to

that in Kumar et al. (2023b) which in turn is a slight modification to that in Dohare et al. (2023).
**Transformer:** We use a decoder model with a single layer consisting of 2 heads, dimension 16 for each head, and with 256 neurons in the feed forward layer with ReLU activations. We deploy this architecture on the Permuted Shakespeare problem using the GPT-2 BPE tokenizer (limited to the set of unique tokens present in the sampled text).

## 3.2 RESULTS AND DISCUSSION

We separately discuss the results for the first four problems (PM, RM, RC, CI) followed by Permuted Shakespeare; we observe additional phenomena in the latter experiment which merit separate discussion.

| Optimizer | SGD | | | | Adam | | | |
|---|---|---|---|---|---|---|---|---|
| Algorithm | PM | RM | RC | CI | PM | RM | RC | CI |
| No Intv. | 0.71 | 0.11 | 0.18 | 0.78 | 0.64 | 0.11 | 0.15 | 0.58 |
| SNR | **0.85** | **0.97** | **0.99** | **0.89** | **0.88** | **0.98** | **0.98** | **0.85** |
| CBP | 0.84 | 0.95 | 0.96 | 0.84 | **0.88** | 0.95 | 0.33 | 0.82 |
| ReDO | 0.83 | 0.72 | 0.98 | 0.87 | 0.85 | 0.67 | 0.74 | 0.80 |
| L2 Reg. | 0.82 | 0.80 | 0.95 | 0.83 | **0.88** | 0.95 | 0.97 | 0.80 |
| L2 Init | 0.83 | 0.91 | 0.97 | 0.83 | **0.88** | 0.96 | **0.98** | 0.83 |
| S&P | 0.83 | 0.92 | 0.97 | 0.85 | **0.88** | 0.96 | 0.97 | 0.81 |
| Layer Norm. | 0.69 | 0.14 | 0.96 | 0.82 | 0.66 | 0.11 | 0.96 | 0.58 |

Table 1: Average accuracy on the last 10% of tasks on the benchmark continual learning problems over 5 seeds. Standard deviations are provided in the extended Table 6.

Table 1 shows that across all four problems (PM, RM, RC, CI) and both SGD and Adam, SNR consistently attains the largest average accuracy on the final 10% of tasks. For each competitor algorithm there is at least one problem on which SNR attains superior accuracy by at least 5 percentage points with SGD and at least 2 percentage points with Adam. We also see that there is no consistent second-best algorithm with the SGD optimizer while L2 Init is consistently the second-best with the Adam optimizer.

Another notable property of SNR is its robustness to the choice of rejection percentile threshold. In contrast, its competitors are not robust to the choice of their hyperparemter(s). On all but RM with SGD, SNR experiences a decrease of at most 2 percentage points in average accuracy when varying its rejection percentile threshold across the range of optimal thresholds found across experiments. On the other hand, increasing the hyperparameter strength by a single order of magnitude, as is common in a hyperparameter sweep, from the optimal value for L2 Init, S&P, and CBP results in a decrease in average accuracy by at least 72 percentage points. See Appendix C for detailed tables.

Next, we turn our attention to the Permuted Shakespeare problem. For the no intervention network with Adam, we see dramatic plasticity loss, as the average loss increases from about 0.15 on the first few tasks to 3.0242 on the last 50 tasks; see Figure 2. In Figure 4 and Figure 3 we see that this plasticity loss is correlated with increasing weight norms in the self-attention and feedforward layers, persistent neuron inactivity, and a collapse in the entropy of the self-attention probabilities.

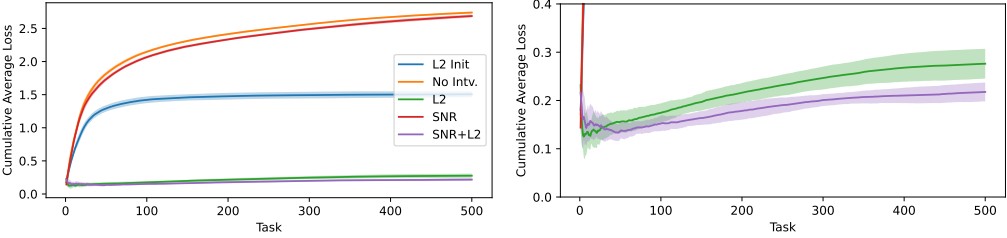

Figure 2: Cumulative average of the loss measured on the final epoch of each task on the Permuted Shakespeare problem. Right panel zooms in to L2 and SNR+L2

| Algorithm | All Tasks | | First 50 Tasks | | Last 50 Tasks | |
|-----------|-----------|--|----------------|--|---------------|--|
| L2 | 0.2762 | (0.0309) | 0.1560 | (0.0236) | 0.3101 | (0.0425) |
| SNR+L2 | 0.2177 | (0.0196) | 0.1370 | (0.0169) | 0.2551 | (0.0454) |
| No Intv. | 2.7397 | (0.0140) | 1.8164 | (0.0295) | 3.0147 | (0.0250) |
| L2 Init | 1.5052 | (0.0437) | 1.2931 | (0.0486) | 1.5262 | (0.0420) |
| SNR | 2.6872 | (0.0222) | 1.7338 | (0.0295) | 3.0242 | (0.0408) |
| CBP | 2.4922 | (0.0171) | 1.3732 | (0.0234) | 2.9410 | (0.0188) |
| L2* | 0.1506 | (0.0279) | 0.1874 | (0.0348) | 0.1092 | (0.0429) |
| SNR+L2* | **0.1402** | (0.0246) | **0.1549** | (0.0107) | **0.0909** | (0.0177) |
| ReDO | 2.3258 | (0.0356) | 1.3689 | (0.0686) | 2.8119 | (0.0373) |

Table 2: Average loss measured on the final epoch of each task with standard deviations over 9 seeds on the Permuted Shakespeare problem. Note, L2* denotes L2 regularization applied only to the attention weights.

We see that resets are by themselves insufficient in mitigating plasticity loss, providing at most a marginal improvement over no intervention. This is unsurprising since neurons are only present in the feedforward layers, unlike the MLP and CNN architectures in the earlier experiments. As such, regularization appears necessary and we see that, over the last 50 tasks, L2 regularization attains an average loss of 0.3101 in contrast to 3.0147 and 3.0242 for no intervention and SNR. This improvement in performance coincides with stable weight norms and non-vanishing average entropy of self-attention probabilities for L2 regularization. In contrast to the earlier problems, L2 Init fares worse than L2 regularization and experiences substantial loss of plasticity, although to a lesser extent than the no intervention network.

While L2 regularization addresses weight blowup, neuron death remains present; see Figure 3. The average loss with L2 increases from 0.1560, over the first 50 tasks, to 0.3101, over the final 50 tasks. This prompts us to consider using SNR in addition to L2 regularization. This largely eliminates neuron death (see the right panel of Figure 3), while stabilizing weight norms and maintaining entropy of self-attention probabilities, providing the lowest loss (0.2551) over the final 50 tasks. As an ablation, we evaluate performance when applying L2 regularization only to the attention weights, which we denote by L2* and SNR+L2* in Table 2.

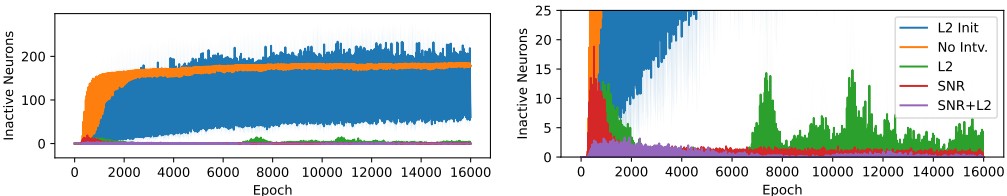

Figure 3: Number of inactive neurons for 5 consecutive training steps on the PS experiment.

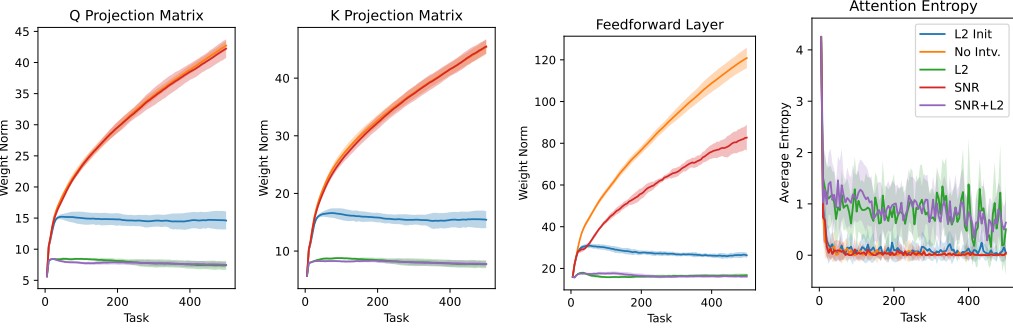

Figure 4: Parameter norms and attention-score entropy on the Permuted Shakespeare experiment.

### 3.2.1 SCALED PERMUTED SHAKESPEARE

While the scale of our Permuted Shakespeare problem serves as a simple benchmark problem for evaluating a series of continual learning algorithms and hyperparameter choices for language models, it is also of interest to investigate the effect of model and dataset scale on plasticity. To this end, we scale the number of non-embedding weights in our transformer network by a factor of $N = 16$, increasing the number of heads to 8 and number of neurons to 1024. In line with scaling laws (Kaplan et al., 2020), we increase the size of our dataset by a factor of $16^{0.74}$, specifically to 254'976 tokens per task. The rest of the problem setup remains unchanged; we train the network for 100 epochs on 500 tasks in sequence. To facilitate the larger token count, we train on a sample of 254'976 tokens worth of text from the complete set of plays by William Shakespeare.

We limit our experiment to 4 random seeds, scales 1 and 16, evaluating only SNR+L2 and L2 with hyperparameters $\eta = 0.05$ and $\lambda = 10^{-4}$, presenting our results in Table 3. We first note that as model and dataset size grow, the gap in average loss between L2-regularization and SNR+L2-regularization grows substantially. Simultaneously, we see a dramatic increase in the proportion of inactive neurons with L2-regularization. At any time step, on average $0.06\%$ of neurons are inactive at scale $N = 1$ while $32.9\%$ are inactive at scale $N = 16$. These results suggest that resets can play a critical role in maintaining plasticity in large-scale language models. See Appendix A which shows a scale 256 experiment, but run only up to 50 tasks, showing similar merits.

| Algorithm | Loss (All Tasks) | | Loss (Last 50 Tasks) | | Dead Neuron Rate |
|---|---|---|---|---|---|
| L2 - Scale 1 | 0.159 | (0.025) | 0.152 | (0.016) | 0.06% |
| SNR+L2 - Scale 1 | 0.154 | (0.008) | 0.141 | (0.031) | 0.00% |
| L2 - Scale 16 | 0.377 | (0.016) | 0.410 | (0.052) | 32.9% |
| SNR+L2 - Scale 16 | 0.324 | (0.028) | 0.332 | (0.073) | $1.41 \cdot 10^{-6}\%$ |

Table 3: Average loss on the final epoch of each task for the scaled Permuted Shakespeare experiments, with means and standard deviations reported over 4 seeds.

### 3.2.2 GENERALIZATION

Lee et al. (2024) recently propose separating training loss from test loss in studying plasticity; the extant literature and the present work focuses largely on measuring training loss on each task. As a complement, we briefly consider measuring test loss on a holdout set for each task. We consider three setups: first, we consider the PM problem, and measure test loss on a set of 10000 image label-pairs for each task; we annotate this setup PM-G1. Our second setup is a modification of PM-G1: in each task we add noise to the labels by randomly re-labeling a fraction of the training images. We decrease the fraction of images with noisy labels from 50% on the first task to 0 on the last; we annotate this task PM-G2. Finally in analogy to PM-G2, we consider a variant of the RC problem (RC-G1) where we decay the fraction of random labels from 50% to 0 linearly across tasks. PM-G2 and RC-G1 are analogous to the setup in Lee et al. (2024).

Detailed results of these experiments are presented in Appendix A.2; we draw the following conclusions: first, SNR displays similar relative merits to competing algorithms as in our main body of experiments on training loss. Second, through a careful study of variants of the setup in PM-G2 where we alter the number of passes through the dataset relevant to each task, we point out an important issue that requires careful attention if one is to study test loss: specifically, confounding the effects of overfitting with plasticity loss.

## 4 THEORETICAL ANALYSIS OF LEARNING A SINGLE RELU

One common hypothesis that seeks to explain plasticity loss is that the network weights obtained from minimizing loss over some task yield poor initializations for a subsequent task; these initializations effectively lead to poor local minima. This section attempts to make this effect transparent through the analysis of the simplest non-trivial task: learning a single ReLU. Vardi et al. (2021) have already established that for random initializations a ReLU can be efficiently learned. Here we ask what happens when the initialization is picked adversarially; one can think of this adversarial initialization as corresponding to optimal weights for an earlier ReLU learning task.

**Theorem 4.1.** [Informal restatement of Theorems E.2 and E.1] There exists a general class of target ReLUs, a distribution over example inputs, and a distribution over initial weights for which:

- Gradient descent with L2 Init or L2 regularization fail to learn the target ReLU with positive probability with respect to the distribution over initial weights. Specifically the average expected MSE exceeds a positive constant as the number of training examples $T$ grows with a probability bounded from below by a positive constant.

- Gradient descent with oracle resets achieves an average expected MSE of $O(1/T)$ with probability one.

The oracle resets in the second result above effectively know precisely whether or not a neuron will or will not fire over the input distribution and are in this sense oracular. Reset methods must come as close as possible to mimicking such an oracle; Propositions 2.1 and 2.2 make the case for why SNR might be a good such candidate and why competing reset proposals may fall short. See Appendix E for a formal setup, statement of results, and proofs.

## 5 DISCUSSION AND LIMITATIONS

A common explanation for plasticity loss is that the network weights obtained from minimizing loss over some task yield poor initializations for a subsequent task – this explanation continues to motivate the vast majority of algorithms that attempt to combat plasticity loss. Specifically, regularization based algorithms can be viewed as regularizing towards a 'good' initial set of weights, whereas reset based algorithms can be viewed as explicitly reinitializing weights to good random initializations.

**Theoretical Motivation:** Our theoretical development showed that regularization methods themselves are insufficient at combatting the plasticity loss problem, at least in the case of learning ReLUs (Theorem 4.1). Further we showed that existing proposals to detect whether it was appropriate to reset a ReLU (based on ad hoc notions of a neuron's utility) could have high type 1 error rates relative to an optimal resetting mechanism (Proposition 2.2). SNR was designed to minimize the error rate in solving this detection problem (Proposition 2.1).

**SOTA Performance:** Across a range of experiments we see that neuron inactivity is an important correlate of plasticity loss, and that SNR consistently outperforms other reset based methods as well as regularization methods (Table 1, Figure 2). We also observed that SNR was robust to its hyperparameters while the adhoc utility schemes employed by other reset methods tended to make them somewhat brittle (Appendix C). Finally, the extant literature has focused on measuring plasticity loss through the lens of training loss, but it is also natural to ask about test loss. We see that SNR enjoys similar relative merits in this context as well (Table 5), but one has to be careful to not confound issues of overfitting with plasticity loss (Figure 6).

**Attention Menchanisms:** Whereas neuron death is an important correlate of plasticity loss, in the case of language models, a crucial component of the architecture – the attention mechanism – does not involve neurons so that the notion of resetting is irrelevant to that component. We have shown that plasticity loss occurs nonetheless and that it is associated with a collapse in the entropy of the attention layer along with neuron death in the feedforward layers (Figures 3 and 4). As such, we see that combining our resets along with L2 regularization of the attention mechanism is important to preserve plasticity (Table 2). Language models also give us a natural substrate to consider the plasticity loss phenomenon vis-a-vis scaling laws, and we see similar relative merits across several orders of magnitude of model scales (Table 3, Appendix 4).

**Limitations:** We recognize a few limitations in the present work. First, whereas we have shown the plasticity loss phenomenon over several orders of model scale our largest models have up to 5M parameters. This is large in the context of the literature on plasticity loss but small in a practical sense. We believe that overcoming this requires understanding ways of exploring this phenomenon without the brute force effort of feeding a model a large sequence of tasks (which is fundamentally serial and hard to accelerate). A second limitation is the theoretical characterization of the phenomenon itself; the approach in the literature has been largely phenomenological – fixing conjectured root causes for what is really an increase in dynamic regret over time. While Section 4 took a first step, future work ought to connect the plasticity loss phenomenon tightly with the optimization landscape.

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

## A  ADDITIONAL EXPERIMENTAL RESULTS

### A.1  EVEN LARGER SCALE EXPERIMENT FOR PERMUTED SHAKESPEARE

We increase the scale further for the Permuted Shakespeare experiment from our original model by a factor of 256, resulting in a model with 5.1 million non-embedding parameters; results are in Table 4. In accordance with scaling laws, we simultaneously increase the number of tokens or examples per task by a factors of $256^{0.74}$, resulting in 1.9 million tokens or 15'500 training examples, with a context window of 128, for the 256-scale model. Due to computational constraints, we reduce the number of tasks from 500 to 50, but keep a training regime of 100 epochs per task. At the largest scale, over the 50 tasks, our model is trained for 100 epochs over 99 million tokens or 775'000 distinct 128-token-long training examples.

The scale of our largest model is comparable to the scale of the largest transformers considered in recent literature on plasticity loss, namely ViT Tiny (5 million parameters) in Lee et al. (2024). The largest dataset on which Lee et al. (2024) train on is the Tiny Imagenet dataset consisting of 100'000 training examples, on which models are trained for 100 epochs; our largest experiment consists of 775'000 training examples. It is important to consider the scale of datasets used in continual learning experiments as the phenomenon of plasticity loss has been shown to be correlated with the amount of data on which a network trains (Kumar et al., 2023a; Dohare et al., 2021).

| Algorithm | Train Loss at Scale 256 | |
|---|---|---|
| SNR+L2 | 0.4576 | (0.0017) |
| L2 | 0.4681 | (0.0025) |

Table 4: Average training loss on the final epoch over all 50 tasks of the 256-scale Permuted Shakespeare problem with standard deviations reported for 5 random seeds.

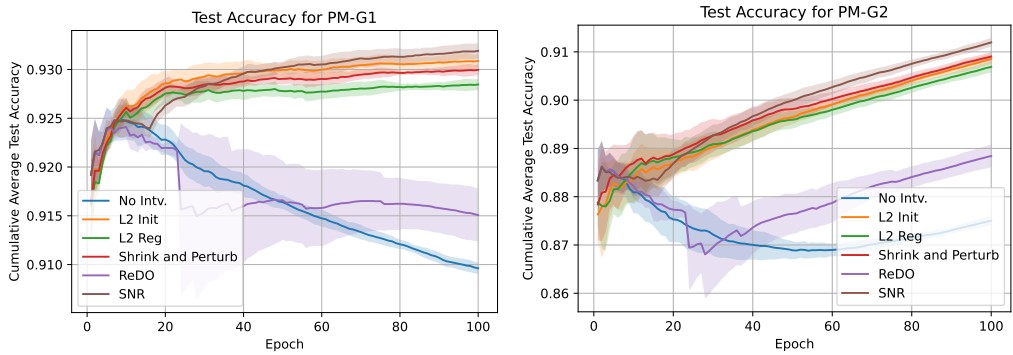

Figure 5: Cumulative average test accuracy for PM-G1 and PM-G2 experiments.

**It is important to note that relative to Table 3 where we show plasticity loss for 500 tasks, the above experiment has only been run to 50 tasks (due to computational budget constraints) and already shows a gap between SNR+L2 and L2.**

A.2  GENERALIZATION RESULTS

Table 5 shows test loss on the last 10% of tasks for PM-G1, PM-G2 and RC-G1. Figure 5 shows the test accuracy for PM-G1 and PM-G2. We see that SNR enjoys similar relative merits to its competitors in PM-G1 and PM-G2. For RC-G1, it appears that there is no plasticity loss and as such all of the different algorithms perform similarly. Notice that this is not contrary to our observations in RC – there the labels across tasks were entirely unrelated.

We consider next a slight tweak on RC-G1. Specifically, each task in RC-G1 consists of 16 epochs; what happens if we do more? The bottom three curves of Figure 6 show the results of doing 100 epochs for each task. We see here that test loss for no intervention and SNR begin to suffer. As it turns out this is misleading – specifically, we see that the test accuracy for all three algorithms is significantly below test accuracy when one does only 16 epochs (as reported in Table 5). That is, the decay in test performance is likely attributable to overfitting in each task, and while L2 overfits as well, it is unsurprisingly more resilient. Put a different way, this overfitting can be controlled by the usual tools to combat overfitting. The top three curves of Figure 6 show that SNR performs equivalently to L2 showing that early stopping would accomplish this comfortably in this context.

| Algorithm | PM-G1 | | PM-G2 | | RC-G1 | |
|---|---|---|---|---|---|---|
| No Intv. | 0.8999 | (0.0041) | 0.8931 | (0.0013) | **0.5526** | (0.0034) |
| L2 Init | 0.9320 | (0.0037) | 0.9284 | (0.0032) | N/A | |
| L2 | 0.9297 | (0.0034) | 0.9258 | (0.0032) | **0.5651** | (0.0023) |
| S&P | 0.9313 | (0.0033) | 0.9281 | (0.0027) | N/A | |
| ReDO | 0.9086 | (0.0028) | 0.9083 | (0.0021) | N/A | |
| SNR | **0.9318** | (0.0001) | **0.9325** | (0.0029) | **0.5650** | (0.0013) |

Table 5: Test accuracy over the last 10% of tasks (mean and standard deviation) for PM-G1, PM-G2, and RC-G1.

B  COMPLETE RESULTS FOR BENCHMARK PROBLEMS

To maintain brevity in the main body of the paper, we include here the complete results for the benchmark problems PM, RC, RM, and CI which include both the mean and standard deviation of terminal task accuracies over random seeds.

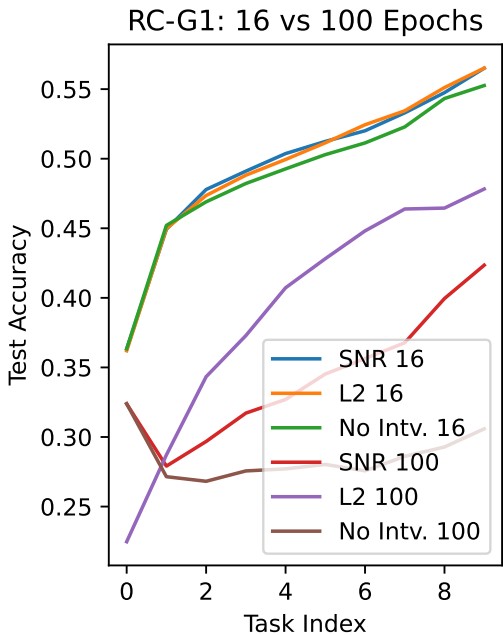

Figure 6: Test accuracy for the RC-G1 experiments evaluated with 16 and 100 epochs per task.

| Optimizer | SGD | | | |
|---|---|---|---|---|
| **Algorithm** | **PM** | **RM** | **RC** | **CI** |
| No Intv. | 0.710 (0.007) | 0.113(0.004) | 0.180 (0.011) | 0.784 (0.019) |
| SNR | **0.851**(0.002) | **0.975** (0.001) | **0.987** (0.002) | **0.888** (0.010) |
| CBP | 0.844(0.002) | 0.951 (0.007) | 0.961 (0.011) | 0.840 (0.015) |
| ReDO | 0.831 (0.013) | 0.716 (0.024) | 0.981 (0.003) | 0.869 (0.038) |
| L2 Reg. | 0.818 (0.001) | 0.803 (0.011) | 0.952 (0.006) | 0.833 (0.011) |
| L2 Init | 0.829 (0.001) | 0.913 (0.001) | 0.966 (0.002) | 0.832 (0.010) |
| S&P | 0.826 (0.002) | 0.920 (0.009) | 0.971 (0.004) | 0.853 (0.006) |
| Layer Norm. | 0.687 (0.009) | 0.143 (0.015) | 0.959 (0.005) | 0.819 (0.009) |
| **Optimizer** | **Adam** | | | |
| **Algorithm** | **PM** | **RM** | **RC** | **CI** |
| No Intv. | 0.641 (0.007) | 0.114 (0.005) | 0.151 (0.005) | 0.581 (0.081) |
| SNR | **0.889**(0.001) | **0.982**(0.001) | **0.976** (0.002) | **0.847** (0.005) |
| CBP | 0.876 (0.001) | 0.948 (0.003) | 0.331 (0.312) | 0.818 (0.005) |
| ReDO | 0.846 (0.002) | 0.671 (0.021) | 0.744 (0.131) | 0.803 (0.063) |
| L2 Reg. | 0.876(0.002) | 0.948 (0.002) | 0.967 (0.011) | 0.803 (0.009) |
| L2 Init | 0.883 (0.002) | 0.961 (0.003) | **0.976** (0.002) | 0.827 (0.008) |
| S&P | 0.876 (0.002) | 0.955 (0.006) | 0.971 (0.005) | 0.814 (0.005) |
| Layer Norm. | 0.662 (0.001) | 0.113 (0.005) | 0.955 (0.005) | 0.651 (0.053) |

Table 6: Average accuracy on the last 10% of tasks on the benchmark continual learning problems with standard deviations over 5 seeds.

## C  ADDITIONAL EXPERIMENTAL DETAILS AND HYPERPARAMETER SWEEP

With SGD we train with learning rate $10^{-2}$ on all problems except Random Label MNIST, for which we train with learning rate $10^{-1}$. With Adam we train with learning rate $10^{-3}$ on all problems, including Permuted Shakespeare and we use the standard parameters of $\beta_1 = 0.9, \beta_2 = 0.999$, and $\epsilon = 10^{-7}$. For Permuted Shakespeare we train our networks solely with Adam. The learning rates were selected after an initial hyperparameter sweep.

For each algorithm we vary its hyperparameter(s) by an appropriate constant over 7 choices, effectively varying the hyperparameters over a log scale. With the exception of the Permuted Shakespeare experiment, we limit over hyperparameter search to 5 choices. In Table 7 we provide the hyperparameter sweep for the 4 benchmark problems. CBP's replacement rate $r$ is to be interpreted as one replacement per layer every $r^{-1}$ training examples, as presented in Dohare et al. (2024). ReDO's reset frequency $r$ determines the frequency of resets in units of tasks, as implemented and evaluated in Kumar et al. (2023b).

| | Hyperparameter Strength | | | | | | |
|---|---|---|---|---|---|---|---|
| Algorithm | 0 | 1 | 2 | 3 | 4 | 5 | 6 |
| L2 Reg. ($\lambda$) | $10^{-6}$ | $10^{-5}$ | $10^{-4}$ | $10^{-3}$ | $10^{-2}$ | $10^{-1}$ | $10^{0}$ |
| L2 Init ($\lambda$) | $10^{-6}$ | $10^{-5}$ | $10^{-4}$ | $10^{-3}$ | $10^{-2}$ | $10^{-1}$ | $10^{0}$ |
| S&P (1-p) | $10^{-8}$ | $10^{-7}$ | $10^{-6}$ | $10^{-5}$ | $10^{-4}$ | $10^{-3}$ | $10^{-2}$ |
| S&P ($\sigma$) | $10^{-6}$ | $10^{-5}$ | $10^{-4}$ | $10^{-3}$ | $10^{-2}$ | $10^{-1}$ | $10^{0}$ |
| CBP ($r$) | $10^{-7}$ | $10^{-6}$ | $10^{-5}$ | $10^{-4}$ | $10^{-3}$ | $10^{-2}$ | $10^{-1}$ |
| ReDO ($\tau$) | 0 | 0.01 | 0.02 | 0.04 | 0.08 | 0.16 | 0.32 |
| ReDO ($r$) | 64 | 32 | 16 | 8 | 4 | 2 | 1 |
| SNR ($\eta$) | 0.08 | 0.04 | 0.02 | 0.01 | 0.005 | 0.0025 | 0.00125 |

Table 7: Hyperparameter sweep for the Permuted MNIST (PM), Random Label MNIST (RM), Random Label CIFAR (RC), and Continual ImageNet (CI) problems.

| | Hyperparameter Strength | | | | |
|---|---|---|---|---|---|
| Algorithm | 0 | 1 | 2 | 3 | 4 |
| L2 Reg. ($\lambda$) | $10^{-6}$ | $10^{-5}$ | $10^{-4}$ | $10^{-3}$ | $10^{-2}$ |
| L2 Init ($\lambda$) | $10^{-6}$ | $10^{-5}$ | $10^{-4}$ | $10^{-3}$ | $10^{-2}$ |
| SNR ($\eta$) | 0.1 | 0.05 | 0.03 | 0.01 | 0.001 |
| SNR + L2 Reg ($\eta$) | 0.1 | 0.05 | 0.03 | 0.01 | 0.001 |

Table 8: Hyperparameter sweep for the Permuted Shakespeare problem. For the combination of SNR and L2 regularization we use the regularization strength of $10^{-4}$, the best performing regularization strength for L2 regularization, and vary the rejection percentile threshold $\eta$

.

| | Hyperparameter Strength | | | | |
|---|---|---|---|---|---|
| Algorithm | 0 | 1 | 2 | 3 | 4 |
| L2 Reg. ($\lambda$) | 1.284 | 0.883 | 0.348 | 0.853 | 4.269 |
| | (0.037) | (0.059) | (0.093) | (0.061) | (0.013) |
| L2 Init ($\lambda$) | 1.791 | 1.481 | 1.641 | 2.429 | 5.598 |
| | (0.056) | (0.050) | (0.067) | (0.124) | (0.030) |
| SNR ($\eta$) | 3.034 | 3.024 | 3.012 | 3.027 | 3.025 |
| | (0.022) | (0.041) | (0.025) | (0.029) | (0.045) |
| SNR + L2 Reg ($\eta$) | 0.315 | 0.276 | 0.325 | 0.293 | 0.269 |
| | (0.084) | (0.078) | (0.068) | (0.065) | (0.025) |

Table 9: Average loss of the final epoch of each task in Permuted Shakespeare, averaged over the final 50 tasks, reported as mean (standard deviation) over 5 random seeds.

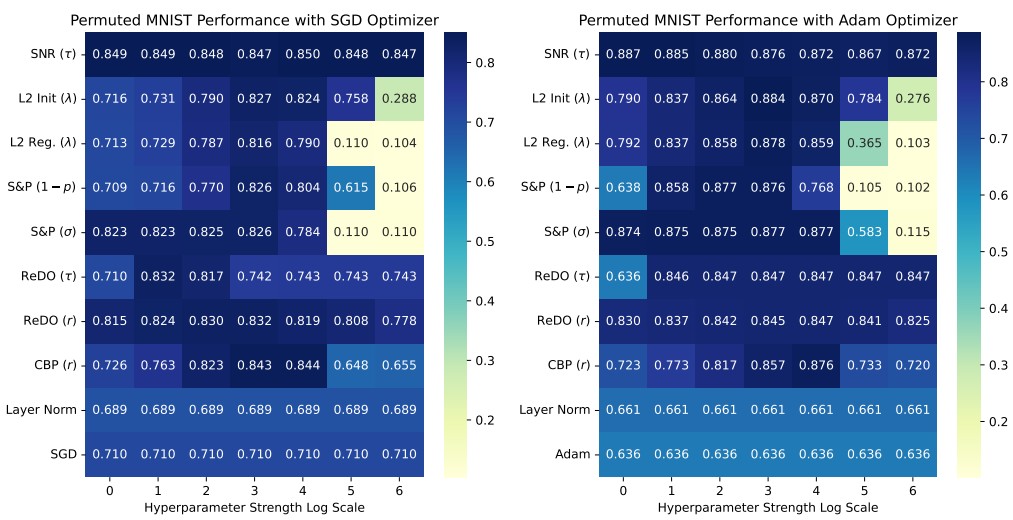

Figure 7: Permuted MNIST hyperparameter sweep results over 5 seeds.

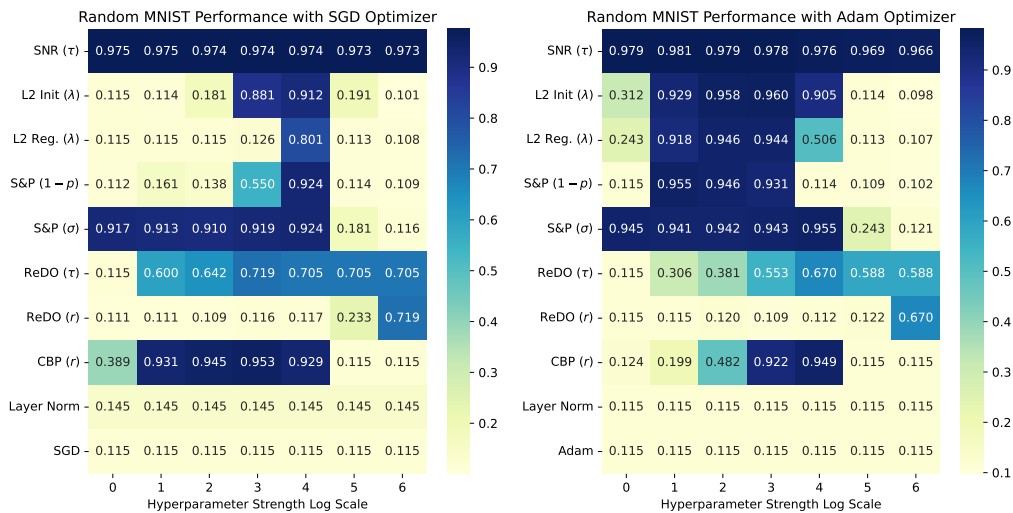

Figure 8: Random LabeL MNIST hyperparameter sweep result over 5 seeds.

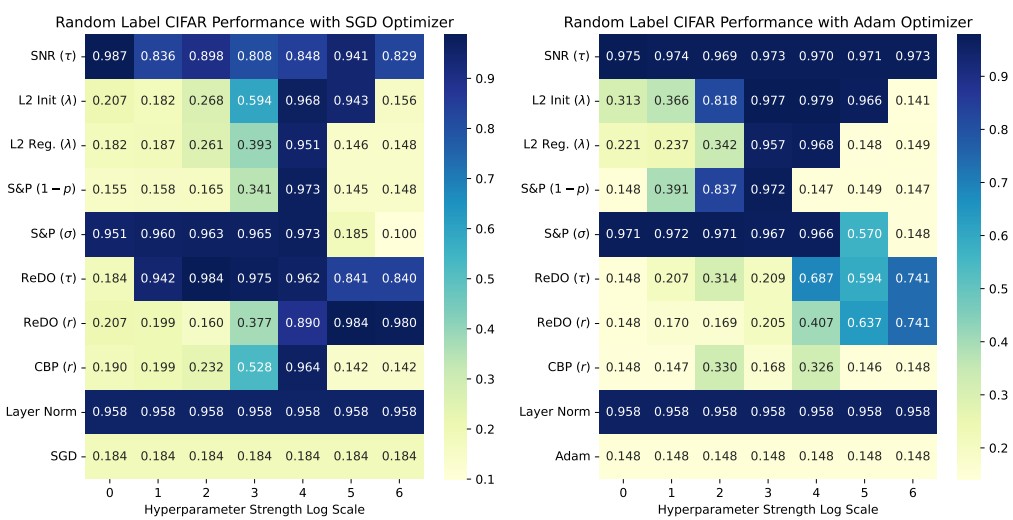

Figure 9: Random Label CIFAR hyperparameter sweep results over 5 seeds.

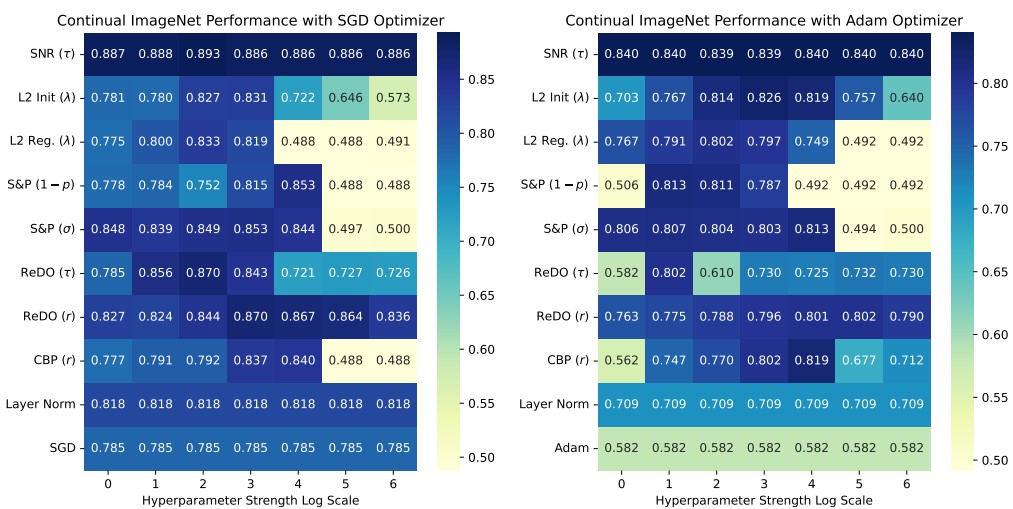

Figure 10: Continual ImageNet hyperparameter sweep results over 5 seeds.

| Algorithm | Hyperparameter Strength Log Scale | | | | | | |
|---|---|---|---|---|---|---|---|
| | 0 | 1 | 2 | 3 | 4 | 5 | 6 |
| SNR ($\eta$) | 0.849 | 0.849 | 0.848 | 0.847 | 0.850 | 0.848 | 0.847 |
| | (0.002) | (0.003) | (0.003) | (0.004) | (0.002) | (0.002) | (0.002) |
| L2 Init ($\lambda$) | 0.716 | 0.731 | 0.790 | 0.827 | 0.824 | 0.758 | 0.288 |
| | (0.001) | (0.001) | (0.001) | (0.002) | (0.001) | (0.001) | (0.017) |
| L2 Reg. ($\lambda$) | 0.713 | 0.729 | 0.787 | 0.816 | 0.790 | 0.110 | 0.104 |
| | (0.002) | (0.002) | (0.001) | (0.002) | (0.003) | (0.003) | (0.002) |
| S&P ($1-p$) | 0.709 | 0.716 | 0.770 | 0.826 | 0.804 | 0.615 | 0.106 |
| | (0.001) | (0.002) | (0.001) | (0.002) | (0.004) | (0.001) | (0.003) |
| S&P ($\sigma$) | 0.823 | 0.823 | 0.825 | 0.826 | 0.784 | 0.110 | 0.110 |
| | (0.001) | (0.002) | (0.002) | (0.002) | (0.002) | (0.002) | (0.002) |
| ReDO ($\tau$) | 0.710 | 0.832 | 0.817 | 0.742 | 0.743 | 0.743 | 0.743 |
| | (0.007) | (0.001) | (0.011) | (0.018) | (0.018) | (0.018) | (0.018) |
| ReDO ($r$) | 0.815 | 0.824 | 0.830 | 0.832 | 0.819 | 0.808 | 0.778 |
| | (0.004) | (0.002) | (0.001) | (0.001) | (0.005) | (0.006) | (0.013) |
| CBP ($r$) | 0.726 | 0.763 | 0.823 | 0.843 | 0.844 | 0.648 | 0.655 |
| | (0.006) | (0.002) | (0.002) | (0.001) | (0.002) | (0.000) | (0.000) |
| Layer Norm | 0.689 | 0.689 | 0.689 | 0.689 | 0.689 | 0.689 | 0.689 |
| | (0.003) | (0.003) | (0.003) | (0.003) | (0.003) | (0.003) | (0.003) |
| SGD | 0.710 | 0.710 | 0.710 | 0.710 | 0.710 | 0.710 | 0.710 |
| | (0.007) | (0.007) | (0.007) | (0.007) | (0.007) | (0.007) | (0.007) |

Table 10: Performance on Permuted MNIST with different hyperparameters reported as mean (standard deviation).

| Algorithm | Hyperparameter Strength Log Scale | | | | | | |
|---|---|---|---|---|---|---|---|
| | 0 | 1 | 2 | 3 | 4 | 5 | 6 |
| SNR ($\eta$) | 0.887 | 0.885 | 0.880 | 0.876 | 0.872 | 0.867 | 0.872 |
| | (0.002) | (0.001) | (0.002) | (0.002) | (0.001) | (0.002) | (0.012) |
| L2 Init ($\lambda$) | 0.790 | 0.837 | 0.864 | 0.884 | 0.870 | 0.784 | 0.276 |
| | (0.001) | (0.002) | (0.002) | (0.002) | (0.002) | (0.001) | (0.008) |
| L2 Reg. ($\lambda$) | 0.792 | 0.837 | 0.858 | 0.878 | 0.859 | 0.365 | 0.103 |
| | (0.002) | (0.002) | (0.002) | (0.002) | (0.004) | (0.008) | (0.002) |
| S&P ($1-p$) | 0.638 | 0.858 | 0.877 | 0.876 | 0.768 | 0.105 | 0.102 |
| | (0.005) | (0.002) | (0.002) | (0.002) | (0.001) | (0.003) | (0.001) |
| S&P ($\sigma$) | 0.874 | 0.875 | 0.875 | 0.877 | 0.877 | 0.583 | 0.115 |
| | (0.002) | (0.002) | (0.002) | (0.002) | (0.002) | (0.003) | (0.001) |
| ReDO ($\tau$) | 0.636 | 0.846 | 0.847 | 0.847 | 0.847 | 0.847 | 0.847 |
| | (0.009) | (0.002) | (0.001) | (0.001) | (0.001) | (0.001) | (0.001) |
| ReDO ($r$) | 0.830 | 0.837 | 0.842 | 0.845 | 0.847 | 0.841 | 0.825 |
| | (0.004) | (0.002) | (0.002) | (0.002) | (0.001) | (0.001) | (0.001) |
| CBP ($r$) | 0.723 | 0.773 | 0.817 | 0.857 | 0.876 | 0.733 | 0.720 |
| | (0.012) | (0.004) | (0.002) | (0.002) | (0.002) | (0.001) | (0.008) |
| Layer Norm | 0.661 | 0.661 | 0.661 | 0.661 | 0.661 | 0.661 | 0.661 |
| | (0.001) | (0.001) | (0.001) | (0.001) | (0.001) | (0.001) | (0.001) |
| Adam | 0.636 | 0.636 | 0.636 | 0.636 | 0.636 | 0.636 | 0.636 |
| | (0.009) | (0.009) | (0.009) | (0.009) | (0.009) | (0.009) | (0.009) |

Table 11: Performance on Permuted MNIST with different hyperparameters reported as mean (standard deviation).

| Algorithm | Hyperparameter Strength Log Scale | | | | | | |
|---|---|---|---|---|---|---|---|
| | 0 | 1 | 2 | 3 | 4 | 5 | 6 |
| SNR ($\eta$) | 0.975 | 0.975 | 0.974 | 0.974 | 0.974 | 0.973 | 0.973 |
| | (0.001) | (0.001) | (0.000) | (0.000) | (0.001) | (0.001) | (0.001) |
| L2 Init ($\lambda$) | 0.115 | 0.114 | 0.181 | 0.881 | 0.912 | 0.191 | 0.101 |
| | (0.006) | (0.009) | (0.027) | (0.017) | (0.006) | (0.008) | (0.010) |
| L2 Reg. ($\lambda$) | 0.115 | 0.115 | 0.115 | 0.126 | 0.801 | 0.113 | 0.108 |
| | (0.006) | (0.006) | (0.006) | (0.013) | (0.010) | (0.007) | (0.009) |
| S&P ($1-p$) | 0.112 | 0.161 | 0.138 | 0.550 | 0.924 | 0.114 | 0.109 |
| | (0.008) | (0.009) | (0.008) | (0.023) | (0.003) | (0.006) | (0.009) |
| S&P ($\sigma$) | 0.917 | 0.913 | 0.910 | 0.919 | 0.924 | 0.181 | 0.116 |
| | (0.005) | (0.004) | (0.007) | (0.004) | (0.003) | (0.005) | (0.006) |
| ReDO ($\tau$) | 0.115 | 0.600 | 0.642 | 0.719 | 0.705 | 0.705 | 0.705 |
| | (0.006) | (0.023) | (0.030) | (0.021) | (0.021) | (0.021) | (0.021) |
| ReDO ($r$) | 0.111 | 0.111 | 0.109 | 0.116 | 0.117 | 0.233 | 0.719 |
| | (0.008) | (0.008) | (0.008) | (0.019) | (0.019) | (0.022) | (0.021) |
| CBP ($r$) | 0.389 | 0.931 | 0.945 | 0.953 | 0.929 | 0.115 | 0.115 |
| | (0.031) | (0.051) | (0.012) | (0.005) | (0.002) | (0.006) | (0.006) |
| Layer Norm | 0.145 | 0.145 | 0.145 | 0.145 | 0.145 | 0.145 | 0.145 |
| | (0.013) | (0.013) | (0.013) | (0.013) | (0.013) | (0.013) | (0.013) |
| SGD | 0.115 | 0.115 | 0.115 | 0.115 | 0.115 | 0.115 | 0.115 |
| | (0.006) | (0.006) | (0.006) | (0.006) | (0.006) | (0.006) | (0.006) |

Table 12: Performance on Random Label MNIST with different hyperparameters reported as mean (standard deviation).

| Algorithm | Hyperparameter Strength Log Scale | | | | | | |
|---|---|---|---|---|---|---|---|
| | 0 | 1 | 2 | 3 | 4 | 5 | 6 |
| SNR ($\eta$) | 0.979 | 0.981 | 0.979 | 0.978 | 0.976 | 0.969 | 0.966 |
| | (0.001) | (0.001) | (0.002) | (0.001) | (0.002) | (0.001) | (0.003) |
| L2 Init ($\lambda$) | 0.312 | 0.929 | 0.958 | 0.960 | 0.905 | 0.114 | 0.098 |
| | (0.050) | (0.011) | (0.007) | (0.001) | (0.003) | (0.004) | (0.009) |
| L2 Reg. ($\lambda$) | 0.243 | 0.918 | 0.946 | 0.944 | 0.506 | 0.113 | 0.107 |
| | (0.038) | (0.008) | (0.009) | (0.004) | (0.043) | (0.007) | (0.009) |
| S&P ($1-p$) | 0.115 | 0.955 | 0.946 | 0.931 | 0.114 | 0.109 | 0.102 |
| | (0.006) | (0.004) | (0.008) | (0.005) | (0.006) | (0.009) | (0.004) |
| S&P ($\sigma$) | 0.945 | 0.941 | 0.942 | 0.943 | 0.955 | 0.243 | 0.121 |
| | (0.009) | (0.009) | (0.007) | (0.013) | (0.004) | (0.015) | (0.006) |
| ReDO ($\tau$) | 0.115 | 0.306 | 0.381 | 0.553 | 0.670 | 0.588 | 0.588 |
| | (0.006) | (0.017) | (0.023) | (0.028) | (0.019) | (0.031) | (0.031) |
| ReDO ($r$) | 0.115 | 0.115 | 0.120 | 0.109 | 0.112 | 0.122 | 0.670 |
| | (0.006) | (0.006) | (0.009) | (0.008) | (0.008) | (0.028) | (0.019) |
| CBP ($r$) | 0.124 | 0.199 | 0.482 | 0.922 | 0.949 | 0.115 | 0.115 |
| | (0.030) | (0.055) | (0.077) | (0.007) | (0.004) | (0.006) | (0.006) |
| Layer Norm | 0.115 | 0.115 | 0.115 | 0.115 | 0.115 | 0.115 | 0.115 |
| | (0.006) | (0.006) | (0.006) | (0.006) | (0.006) | (0.006) | (0.006) |
| Adam | 0.115 | 0.115 | 0.115 | 0.115 | 0.115 | 0.115 | 0.115 |
| | (0.006) | (0.006) | (0.006) | (0.006) | (0.006) | (0.006) | (0.006) |

Table 13: Performance on Random Label MNIST with different hyperparameters reported as mean (standard deviation).

| Algorithm | Hyperparameter Strength Log Scale | | | | | | |
|---|---|---|---|---|---|---|---|
| | 0 | 1 | 2 | 3 | 4 | 5 | 6 |
| SNR ($\eta$) | 0.887 | 0.888 | 0.893 | 0.886 | 0.886 | 0.886 | 0.886 |
| | (0.009) | (0.007) | (0.006) | (0.004) | (0.004) | (0.004) | (0.004) |
| L2 Init ($\lambda$) | 0.781 | 0.780 | 0.827 | 0.831 | 0.722 | 0.646 | 0.573 |
| | (0.028) | (0.016) | (0.011) | (0.008) | (0.014) | (0.014) | (0.018) |
| L2 Reg. ($\lambda$) | 0.775 | 0.800 | 0.833 | 0.819 | 0.488 | 0.488 | 0.491 |
| | (0.010) | (0.013) | (0.012) | (0.007) | (0.000) | (0.000) | (0.001) |
| S&P ($1-p$) | 0.778 | 0.784 | 0.752 | 0.815 | 0.853 | 0.488 | 0.488 |
| | (0.023) | (0.024) | (0.073) | (0.010) | (0.008) | (0.000) | (0.000) |
| S&P ($\sigma$) | 0.848 | 0.839 | 0.849 | 0.853 | 0.844 | 0.497 | 0.500 |
| | (0.002) | (0.010) | (0.008) | (0.008) | (0.010) | (0.005) | (0.000) |
| ReDO ($\tau$) | 0.785 | 0.856 | 0.870 | 0.843 | 0.721 | 0.727 | 0.726 |
| | (0.024) | (0.012) | (0.010) | (0.015) | (0.041) | (0.043) | (0.040) |
| ReDO ($r$) | 0.827 | 0.824 | 0.844 | 0.870 | 0.867 | 0.864 | 0.836 |
| | (0.027) | (0.038) | (0.029) | (0.010) | (0.016) | (0.011) | (0.009) |
| CBP ($r$) | 0.777 | 0.791 | 0.792 | 0.837 | 0.840 | 0.488 | 0.488 |
| | (0.022) | (0.016) | (0.013) | (0.012) | (0.008) | (0.000) | (0.000) |
| Layer Norm | 0.818 | 0.818 | 0.818 | 0.818 | 0.818 | 0.818 | 0.818 |
| | (0.011) | (0.011) | (0.011) | (0.011) | (0.011) | (0.011) | (0.011) |
| SGD | 0.785 | 0.785 | 0.785 | 0.785 | 0.785 | 0.785 | 0.785 |
| | (0.024) | (0.024) | (0.024) | (0.024) | (0.024) | (0.024) | (0.024) |

Table 14: Performance on Continual ImageNet with different hyperparameters reported as mean (standard deviation).

| Algorithm | Hyperparameter Strength Log Scale | | | | | | |
|---|---|---|---|---|---|---|---|
| | 0 | 1 | 2 | 3 | 4 | 5 | 6 |
| SNR ($\eta$) | 0.840 | 0.840 | 0.839 | 0.839 | 0.840 | 0.840 | 0.840 |
| | (0.002) | (0.005) | (0.006) | (0.006) | (0.006) | (0.006) | (0.006) |
| L2 Init ($\lambda$) | 0.703 | 0.767 | 0.814 | 0.826 | 0.819 | 0.757 | 0.640 |
| | (0.052) | (0.025) | (0.005) | (0.009) | (0.005) | (0.011) | (0.014) |
| L2 Reg. ($\lambda$) | 0.767 | 0.791 | 0.802 | 0.797 | 0.749 | 0.492 | 0.492 |
| | (0.018) | (0.013) | (0.012) | (0.009) | (0.021) | (0.001) | (0.000) |
| S&P ($1-p$) | 0.506 | 0.813 | 0.811 | 0.787 | 0.492 | 0.492 | 0.492 |
| | (0.025) | (0.010) | (0.008) | (0.008) | (0.001) | (0.000) | (0.000) |
| S&P ($\sigma$) | 0.806 | 0.807 | 0.804 | 0.803 | 0.813 | 0.494 | 0.500 |
| | (0.012) | (0.004) | (0.011) | (0.007) | (0.010) | (0.000) | (0.001) |
| ReDO ($\tau$) | 0.582 | 0.802 | 0.610 | 0.730 | 0.725 | 0.732 | 0.730 |
| | (0.075) | (0.018) | (0.120) | (0.012) | (0.016) | (0.016) | (0.013) |
| ReDO ($r$) | 0.763 | 0.775 | 0.788 | 0.796 | 0.801 | 0.802 | 0.790 |
| | (0.033) | (0.018) | (0.013) | (0.007) | (0.008) | (0.018) | (0.006) |
| CBP ($r$) | 0.562 | 0.747 | 0.770 | 0.802 | 0.819 | 0.677 | 0.712 |
| | (0.087) | (0.032) | (0.014) | (0.004) | (0.003) | (0.092) | (0.039) |
| Layer Norm | 0.709 | 0.709 | 0.709 | 0.709 | 0.709 | 0.709 | 0.709 |
| | (0.014) | (0.014) | (0.014) | (0.014) | (0.014) | (0.014) | (0.014) |
| Adam | 0.582 | 0.582 | 0.582 | 0.582 | 0.582 | 0.582 | 0.582 |
| | (0.075) | (0.075) | (0.075) | (0.075) | (0.075) | (0.075) | (0.075) |

Table 15: Performance on Continual ImageNet with different hyperparameters reported as mean (standard deviation).

| Algorithm | Hyperparameter Strength Log Scale | | | | | | |
|---|---|---|---|---|---|---|---|
| | 0 | 1 | 2 | 3 | 4 | 5 | 6 |
| SNR ($\eta$) | 0.975 | 0.974 | 0.969 | 0.973 | 0.970 | 0.971 | 0.973 |
| | (0.003) | (0.001) | (0.006) | (0.004) | (0.004) | (0.004) | (0.002) |
| L2 Init ($\lambda$) | 0.313 | 0.366 | 0.818 | 0.977 | 0.979 | 0.966 | 0.141 |
| | (0.076) | (0.096) | (0.165) | (0.005) | (0.003) | (0.004) | (0.006) |
| L2 Reg. ($\lambda$) | 0.221 | 0.237 | 0.342 | 0.957 | 0.968 | 0.148 | 0.149 |
| | (0.049) | (0.029) | (0.035) | (0.012) | (0.002) | (0.004) | (0.004) |
| S&P ($1-p$) | 0.148 | 0.391 | 0.837 | 0.972 | 0.147 | 0.149 | 0.147 |
| | (0.004) | (0.228) | (0.170) | (0.002) | (0.005) | (0.004) | (0.005) |
| S&P ($\sigma$) | 0.971 | 0.972 | 0.971 | 0.967 | 0.966 | 0.570 | 0.148 |
| | (0.004) | (0.002) | (0.002) | (0.009) | (0.004) | (0.039) | (0.008) |
| ReDO ($\tau$) | 0.148 | 0.207 | 0.314 | 0.209 | 0.687 | 0.594 | 0.741 |
| | (0.004) | (0.115) | (0.330) | (0.079) | (0.125) | (0.089) | (0.128) |
| ReDO ($r$) | 0.148 | 0.170 | 0.169 | 0.205 | 0.407 | 0.637 | 0.741 |
| | (0.004) | (0.044) | (0.045) | (0.077) | (0.165) | (0.155) | (0.128) |
| CBP ($r$) | 0.148 | 0.147 | 0.330 | 0.168 | 0.326 | 0.146 | 0.148 |
| | (0.004) | (0.007) | (0.302) | (0.026) | (0.074) | (0.006) | (0.004) |
| Layer Norm | 0.958 | 0.958 | 0.958 | 0.958 | 0.958 | 0.958 | 0.958 |
| | (0.006) | (0.006) | (0.006) | (0.006) | (0.006) | (0.006) | (0.006) |
| Adam | 0.148 | 0.148 | 0.148 | 0.148 | 0.148 | 0.148 | 0.148 |
| | (0.004) | (0.004) | (0.004) | (0.004) | (0.004) | (0.004) | (0.004) |

Table 16: Performance on Random Label CIFAR with different hyperparameters reported as mean (standard deviation).

| Algorithm | Hyperparameter Strength Log Scale | | | | | | |
|---|---|---|---|---|---|---|---|
| | 0 | 1 | 2 | 3 | 4 | 5 | 6 |
| SNR ($\eta$) | 0.987 | 0.836 | 0.898 | 0.808 | 0.848 | 0.941 | 0.829 |
| | (0.002) | (0.242) | (0.112) | (0.124) | (0.183) | (0.064) | (0.173) |
| L2 Init ($\lambda$) | 0.207 | 0.182 | 0.268 | 0.594 | 0.968 | 0.943 | 0.156 |
| | (0.040) | (0.012) | (0.068) | (0.135) | (0.004) | (0.007) | (0.005) |
| L2 Reg. ($\lambda$) | 0.182 | 0.187 | 0.261 | 0.393 | 0.951 | 0.146 | 0.148 |
| | (0.012) | (0.024) | (0.097) | (0.265) | (0.005) | (0.004) | (0.004) |
| S&P ($1-p$) | 0.155 | 0.158 | 0.165 | 0.341 | 0.973 | 0.145 | 0.148 |
| | (0.005) | (0.007) | (0.009) | (0.081) | (0.003) | (0.005) | (0.004) |
| S&P ($\sigma$) | 0.951 | 0.960 | 0.963 | 0.965 | 0.973 | 0.185 | 0.100 |
| | (0.007) | (0.006) | (0.004) | (0.003) | (0.003) | (0.068) | (0.014) |
| ReDO ($\tau$) | 0.184 | 0.942 | 0.984 | 0.975 | 0.962 | 0.841 | 0.840 |
| | (0.013) | (0.054) | (0.004) | (0.020) | (0.038) | (0.039) | (0.057) |
| ReDO ($r$) | 0.207 | 0.199 | 0.160 | 0.377 | 0.890 | 0.984 | 0.980 |
| | (0.057) | (0.067) | (0.025) | (0.220) | (0.121) | (0.004) | (0.001) |
| CBP ($r$) | 0.190 | 0.199 | 0.232 | 0.528 | 0.964 | 0.142 | 0.142 |
| | (0.021) | (0.024) | (0.058) | (0.266) | (0.012) | (0.006) | (0.006) |
| Layer Norm | 0.958 | 0.958 | 0.958 | 0.958 | 0.958 | 0.958 | 0.958 |
| | (0.008) | (0.008) | (0.008) | (0.008) | (0.008) | (0.008) | (0.008) |
| SGD | 0.184 | 0.184 | 0.184 | 0.184 | 0.184 | 0.184 | 0.184 |
| | (0.013) | (0.013) | (0.013) | (0.013) | (0.013) | (0.013) | (0.013) |

Table 17: Performance on Random Label CIFAR with different hyperparameters reported as mean (standard deviation).

# D    PROOFS OF PROPOSITION OF 2.1 AND 2.2

**Proposition D.1** (Restatement of Proposition 2.1). Let $T$ be the $1 - \frac{\lambda}{p-\lambda}$ percentile of a Geometric$(p)$ distribution. Then the optimal hypothesis test takes the form $X_\tau = \mathbf{1}\{Z_\tau = 0\}$ where $\tau = \min(s : Z_s = 1) \wedge T$.

*Proof.* We begin by assuming equal priors $\mathbb{P}(H_0) = \mathbb{P}(H_1) = \frac{1}{2}$. We note that for any time $s$, if $Z_s = 1$ then any optimal hypothesis test must declare $X_s = 0$ as $Z_s = 1$ is impossible under $H_1$ and waiting to make a future declaration will incur additional cost of at least $\lambda$. Therefore, it remains for us to derive an optimal stopping time for the collection of states $\{Z_1 = \ldots = Z_s = 0 : s \in \mathbb{Z}_+\}$.

Let $V(s)$ be the expected total future cost at time $s$ given that we have observed $Z_1 = \ldots = Z_s = 0$. We define

$$
\begin{aligned}
\pi_s &= \mathbb{P}(H_0 | Z_1 = \ldots = Z_s = 0) \\
&= \frac{\mathbb{P}(H_0, Z_1 = \ldots = Z_s = 0)}{\mathbb{P}(Z_1 = \ldots = Z_s = 0)} \\
&= \frac{(1-p)^s \mathbb{P}(H_0)}{(1-p)^s \mathbb{P}(H_0) + \mathbb{P}(H_1)} \\
&= \frac{(1-p)^s}{(1-p)^s + 1} \qquad\qquad\qquad \text{by } \mathbb{P}(H_0) = \mathbb{P}(H_1)
\end{aligned}
$$

If we stop at time $s$ and make a declaration, we choose the hypothesis with higher positive probability in order tom minimize the error probability

$$
\mathbb{P}(X_s = 1 | H_0) + \mathbb{P}(X_s = 0 | H_1)
$$

Thus, the expected cost of stopping is

$$
C^{\text{stop}}(s) = \min\{\pi_s, 1 - \pi_s\}
$$

We can simplify this further by noting that $\pi_s \leq 1 - \pi_s$. We note that

$$
\frac{1}{2}(1-p)^s \leq \frac{1}{2}
$$

by $1 - p \in [0, 1]$. This is equivalent to

$$
(1-p)^s \leq \frac{1}{2}((1-p)^s + 1)
$$

by adding $\frac{1}{2}(1-p)^s$, which in turn, is equivalent to

$$
\pi_s = \frac{(1-p)^s}{(1-p)^s + 1} \leq \frac{1}{2}
$$

Therefore, $\pi_s \leq \frac{1}{2} \leq 1 - \pi_s$ and so we have that

$$
C^{\text{stop}}(s) = \min\{\pi_s, 1 - \pi_s\} = \pi_s
$$

This also implies that if we are to stop at some state $\{Z_1 = \ldots = Z_s = 0\}$, it is optimal to declare $X_s = 1$.

If we continue at time $s$ to $s+1$, we incur an additional delay cost of $\lambda$, and the expected future cost depending on whether we see a $Z_{s+1} = 1$ or $Z_{s+1} = 0$.

- With probability $p\pi_s$ we obserbes $Z_{s+1}$, under $H_0$, and we stop the process with $X_{s+1} = 0$, incurring zero error cost since $Z_{s+1} = 1$ cannot occur under $H_1$.

- With probability $(1-p)\pi_s + (1 - \pi) = 1 - p\pi_s$ we observe $Z_{s+1} = 0$ and the process continues.

Therefore, the expected cost of continuing at time $s$ is

$$C^{\text{cont}}(s) = \lambda + (1 - p\pi_s)V(s + 1)$$

Then the Bellman equation for the optimal cost-to-go function is

$$V(s) = min\{C^{\text{stop}}(s), C^{\text{cont}}(s)\}$$

To determine an optimal stopping time, our goal is to find smallest $T$ for which

$$C^{\text{stop}}(T) \leq C^{\text{cont}}(T) \tag{1}$$

Assuming we stop at time $T$,

$$V(T + 1) = C^{\text{stop}}(T + 1) = \pi_{T+1}$$

Therefore,

$$C^{\text{cont}}(T) = \lambda + (1 - p\pi_s)V(T + 1) = \lambda + (1 - p\pi_s)\pi_{T+1}$$

and to establish (1) it suffices to show that

$$\pi_T \leq \lambda + (1 - p\pi_T)\pi_{T+1} \tag{2}$$

First, we write $\pi_{T+1}$ in terms of $\pi_T$. Under the updating rule for the posterior probability, we have that

$$\pi_{T+1} - \mathbb{P}(H_0|Z_1 = \ldots = Z_{T+1} = 0)$$
$$= \frac{\mathbb{P}(Z_{T+1} = 0|H_0)\mathbb{P}(H_0|Z_1 = \ldots = Z_T = 0)}{\mathbb{P}(Z_{T+1} = 0|Z_1 = \ldots = Z_T = 0)}$$
$$= \frac{(1 - p)\pi_T}{\mathbb{P}(Z_{T+1} = 0|H_0)\pi_T + \mathbb{P}(Z_{T+1} = 0|H_1)(1 - \pi_T)}$$
$$= \frac{(1 - p)\pi_T}{(1 - p)\pi_T + (1 - \pi_T)}$$
$$= \frac{(1 - p)\pi_T}{1 - p\pi_T}$$

Returning to (2), we need to show that

$$\pi_T \leq \lambda + (1 - p\pi_T)\frac{(1 - p)\pi_T}{1 - p\pi_T} = \lambda + (1 - p)\pi_T$$

Simplifying the above inequality, we have that

$$\pi_T \leq \frac{\lambda}{p}$$

Substituting in our formula for $\pi_T$, the above is equivalent to

$$\frac{(1 - p)^\top}{(1 - p)^\top + 1} \leq \frac{\lambda}{p}$$

which after simplification is equivalent to

$$(1 - p)^\top \leq \frac{\frac{\lambda}{p}}{1 - \frac{\lambda}{p}} = \frac{\lambda}{p - \lambda}$$

Let $F$ be the CDF of the Geometric$(p)$ distribution. Let $T^*$ be the $1 - \frac{\lambda}{p-\lambda}$ percentile of the Geometric$(p)$ distribution. Note, since $\lambda < \frac{p}{2}$ then $1 - \frac{\lambda}{p-\lambda} \in (0, 1)$ and is a valid percentile. Then for any $T \geq T^*$ we have that

$$\begin{aligned}
1 - (1 - p)^\top = F(T) \\
\geq F(T^*) \qquad\qquad &\text{by } T \geq T^* \\
\geq 1 - \frac{\lambda}{p - \lambda} \qquad\qquad &\text{by choice of } T^*
\end{aligned}$$

which is equivalent to

$$(1-p)^\top \le \frac{\lambda}{p-\lambda}$$

and therefore, the optimal hypothesis test is to declare $X_T = 1$ for any $T \ge T^*$ if $Z_1 = \ldots = Z_T = 0$. Hence, the optimal hypothesis takes the form of

$$X_\tau = \mathbf{1}\{Z_\tau = 0\} \text{ where } \tau = \min(s : Z_s = 1) \wedge T^*$$

$\square$

**Proposition D.2.** Restatement of Proposition 2.2] The ratio of total error rate with a fixed threshold $r^*$ to that under SNR scales like

$$\Omega\left(\exp\left(\log(\alpha(1-\alpha)^{-1})\left(-\frac{1}{2} + \frac{1}{2}\frac{\log(1-p_1)}{\log(1-p_2)}\right)\right)\right)$$

*Proof.* For notational convenience, define $\bar{\alpha} = \alpha(1-\alpha)^{-1}$. Notice that by Proposition 2.1, under SNR, neuron $i$, $(I = 1, 2)$, is reset if it inactive for any longer that time $\bar{T} = \log(\bar{\alpha}))/\log(1-p_i)$. Consequently, the expected delay penalty, $\mathbb{E}[\tau|H_0] + \mathbb{E}[\tau|H_1]$ for neuron $i$, is simply

$$\frac{\log(\bar{\alpha})}{\log(1-p_i)} + \frac{1}{p_i}(1-\bar{\alpha})$$

Letting the optimal fixed threshold be $r^*$, we must have that the expected total delay across both neurons under this fixed threshold is at least $2r^*$. This total expected delay can be no larger than that under SNR. Thus,

$$2r^* \le \log(\bar{\alpha})\left(\frac{1}{\log(1-p_1)} + \frac{1}{\log(1-p_2)}\right) + (1-\bar{\alpha})\left(\frac{1}{p_1} + \frac{1}{p_2}\right)$$

But the sum of the error rates across the two neurons with the fixed threshold $r^*$ is at least $(1-p_1)^{r^*} + \bar{\alpha}$, while the total error rate under SNR is precisely $\bar{\alpha}$. Dividing these two quantities and employing the upper bound derived on $r^*$ then yields the result. $\square$

## E LEARNING A SINGLE ReLU

In this section we prove Theorem 4.1.

### E.1 PRELIMINARIES

We aim to learn a single ReLU-activated neuron with bias, or equivalently, the mapping $f_v : \mathbb{R}^2 \to \mathbb{R}_+$ which we define as

$$f_v(x) = \sigma(v^\top x) = \begin{cases} v^\top x & \text{if } v^\top x \ge 0 \\ 0 & \text{if } v^\top x < 0 \end{cases}$$

We refer to $v = (\tilde{v}, b_v)$ as the *target parameters* where $\tilde{v}$ is the slope and $b_v$ is the bias of the linear map $x \mapsto v^\top x$. Likewise, we denote our model's parameters as $w = (\tilde{w}, b_w)$ where $\tilde{w}$ is the slope and $b_w$ is the bias.

We sample data $x = (\tilde{x}, 1) \sim \text{Uniform}(-L, L) \times \{1\}$ such that the first coordinate $\tilde{x}$ is sampled uniformly from the domain $[-L, L]$ and the second coordinate is a constant 1 so as to model the bias term in a ReLU-activated neuron. We learn the target neuron with respect to the squared loss and we thus define the loss to be

$$F(w) = \mathbb{E}_x[\frac{1}{2}(\sigma(w^\top x) - \sigma(v^\top x))^2] \tag{3}$$

Then the gradient of $F$ is simply

$$\nabla F(w) = \mathbb{E}_x[(\sigma(w^\top x) - \sigma(v^\top x))\mathbb{I}(w^\top x \ge 0)x] \tag{4}$$

where the above expectations are taken with respect to $x \sim \text{Uniform}(-L, L) \times \{1\}$. We minimize $F$ using gradient descent with a constant learning rate $\eta$

$$w_{t+1} = w_t - \eta\nabla F(w_t) \tag{5}$$

We define the regret of learning a single ReLU with an iterative algorithm as

$$\mathbb{E}[R_T] = \mathbb{E}_{w_0}\Big[\sum_{t=0}^{T-1} F(w_t)\Big]$$

where the randomness is over the initialization of $w_0$ and the potential reinitialization of $w_t$ due to resets. Then the average regret is simply

$$\frac{1}{T}\mathbb{E}[R_T] = \frac{1}{T}\mathbb{E}_{w_0}\Big[\sum_{t=0}^{T-1} F(w_t)\Big]$$

For some constant $\tilde{v}_{\max} > 1$ we suppose that $\tilde{v} \in [1, \tilde{v}_{\max}]$ and $b_v \in [-\frac{L}{6}, 0]$. While for some constant $\tilde{w}_{\min} < 1$ we sample $\tilde{w}_0$ uniformly from $[-1, -\tilde{w}_0) \cup (\tilde{w}_0, 1]$ and set $b_{w_0} = 0$, as is customary to initialize neurons with zero bias.

### E.1.1 L2 INIT AND L2 REGULARIZATION

Given a regularization strength $\lambda \geq 0$, we consider the loss with L2 Init regularization as

$$\bar{F}(w) = \mathbb{E}_x\Big[\frac{1}{2}(\sigma(w^\top x) - \sigma(v^\top x))^2\Big] + \frac{\lambda}{2}||w - w_0||^2 = F(w) + \frac{\lambda}{2}||w - w_0||^2$$

Then the gradient of $\bar{F}$ is simply

$$\nabla F(w) = \mathbb{E}_x[(\sigma(w^\top x) - \sigma(v^\top x))\mathbb{I}(w^\top x \geq 0)x] + \lambda(w - w_0) = \nabla F(w) + \lambda(w - w_0)$$

Then the L2 Init gradient descent update is simply

$$w_{t+1} = w_t - \eta\nabla\bar{F}(w_t) = w_t - \eta\nabla F(w_t) - \eta\lambda(w_t - w_0) \tag{6}$$

Similarly, we can consider vanilla L2 regularization whose update is simply

$$w_{t+1} = w_t - \eta\nabla F(w_t) - \eta\lambda w_t \tag{7}$$

Note, if $\lambda = 0$ then we simply retain the update of unregularized gradient descent (5). Then for any sufficiently small learning rate $\eta$ we attain non-vanishing average regret.

**Theorem E.1.** Suppose that $x$ is sampled according to $x \sim \text{Uniform}(-L, L) \times \{1\} \subseteq \mathbb{R}^2$ and that the target parameters $v = (\tilde{v}, b_v)$ satisfy $\tilde{v} > 0$ and $b_v \leq 0$. Then applying gradient descent with L2 Init (6) or L2 regularization (7), with regularization strength $\lambda \geq 0$ and learning rate $\eta > 0$ such that

$$\eta \leq \frac{1}{L^2 + 1}$$

$$\eta\lambda < 1$$

and with $w_0$ sampled uniformly from $([-1, -\tilde{w}_{\min}) \cup (\tilde{w}_{\min}, 1]) \times \{0\}$, for any $\tilde{w}_{\min} > 0$, then with probability $\frac{1}{2}$ over random initializations of $w_0$, the average regret is non-vanishing

$$\frac{1}{T}R_T \geq F(0)$$

*Proof.* The proof follows immediately by Lemma E.4 for L2 Init. A trivial modification of Lemma E.4 yields an identical result for L2 regularization, which we omit for brevity. $\square$

### E.1.2 GRADIENT DESCENT WITH RESETS

For any reset threshold $\epsilon > 0$ we define a reset oracle $\mathcal{O}_\epsilon$ such that for any $w \in \mathbb{R}^2$

$$\mathcal{O}_\epsilon(w) = \begin{cases} \text{True} & \text{if } \sup_{x \in [-L, L] \times \{1\}} w^\top x \leq \epsilon \\ \text{False} & \text{if } \sup_{x \in [-L, L] \times \{1\}} w^\top x > \epsilon \end{cases}$$

We consider the following gradient descent updates with resets

$$u_{t+1} = w_t - \eta\nabla F(w_t) \tag{8}$$

$$w_{t+1} = \begin{cases} u_{t+1} & \text{if } \mathcal{O}_\epsilon(u_{t+1}) = \text{False} \\ \text{sample from Uniform}\left(([-1, -\tilde{w}_{\min}) \cup (\tilde{w}_{\min}, 1]) \times \{0\}\right) & \text{if } \mathcal{O}_\epsilon(u_{t+1}) = \text{True} \end{cases} \tag{9}$$

**Theorem E.2.** Suppose that $x$ is sampled according to $x \sim \text{Uniform}(-L, L) \times \{1\} \subseteq \mathbb{R}^2$ and the target parameters $v = (\tilde{v}, b_v)$ satisfy $\tilde{v} \in [1, \tilde{v}_{\max}]$ and $-\frac{L}{6} \leq b_v \leq 0$ and we denote $v_{\max} = (\tilde{v}_{\max}, -\frac{L}{6})$. Let $\epsilon > 0$ be a reset threshold and suppose that the initial parameters $w_0$ are sampled uniformly from $\tilde{w}_0 \sim [-1, -\tilde{w}_{\min}) \cup (\tilde{w}_{\min}, 1]$ and $b_{w_0} = 0$ such that

$$\tilde{w}_{\min} > \frac{12 \tilde{v}_{\max} \epsilon}{L}$$

and

$$\frac{5}{2} L \geq \epsilon.$$

Then gradient descent with a constant learning rate of

$$\eta \leq \frac{\tilde{w}_{\min}^3 L^6}{3 \cdot 12^2 \cdot 24^3 (2 + ||v_{\max}||)^5 (L^2 + 1)^6}$$

and with resets, i.e. (8) and (9), attains an average regret of

$$\frac{1}{T} \mathbb{E}[R_T] \leq \frac{C}{T}$$

where

$$C = \lceil \frac{32 L^2 F((-1, 0))}{\eta \epsilon^4} \rceil F((-1, 0)) + \frac{3}{2\eta^2} \left( (\tilde{v}_{\max} - \tilde{w}_{\min})^2) + (\frac{L}{6})^2 \right)$$

*Proof.* The theorem is restated and proven as Theorem E.3. $\square$

### E.2 PROOFS

#### E.2.1 PROPERTIES OF THE LOSS FUNCTION

**Lemma E.1.** The ReLU activation function $\sigma(x) = \max\{0, x\}$ is 1-Lipschitz continuous.

*Proof.* Let $x, y \in \mathbb{R}$ be arbitrary. Without loss of generality we suppose that $x \geq y$. We consider two cases. Firstly, if $x \geq 0$ then we have that

$$
\begin{aligned}
|\sigma(x) - \sigma(y)| &= \sigma(x) - \sigma(y) && \text{by } x \geq y \\
&= x - \sigma(y) && \text{by } x \geq 0 \\
&\leq x - y && \text{by } \sigma(y) \geq y \\
&\leq |x - y|
\end{aligned}
$$

As for the case of $x < 0$ then we likewise have that $y < 0$ and so

$$|\sigma(x) - \sigma(y)| = 0 \leq |x - y|$$

Thus, it follows that $\sigma(\cdot)$ is 1-Lipschitz continuous. $\square$

**Lemma E.2.** Suppose that $x$ is sampled according to $x \sim \text{Uniform}(-L, L) \times \{1\} \subseteq \mathbb{R}^2$ and that the target parameters $v = (\tilde{v}, b_v)$ satisfy $\tilde{v} > 0$ and $b_v \leq 0$. Then for $w \leq 0$, $\nabla F(w) = \mathbb{E}_x[\sigma(w^\top x) x]$.

*Proof.* For almost every $\tilde{x} > 0$, $w^\top x = \tilde{w}\tilde{x} + b_w < 0$ and so

$$
\begin{aligned}
(\sigma(w^\top x) - \sigma(v^\top x)) \mathbb{I}(w^\top x \geq 0) x &= 0 && \text{by } \mathbb{I}(w^\top x \geq 0) = 0 \\
&= \sigma(w^\top x) x && \text{by } \sigma(w^\top x) = 0 \text{ since } w^\top x < 0
\end{aligned}
$$

As for $\tilde{x} < 0$, we have that $v^\top x < 0$ since $\tilde{v} > 0$ and $b_v \leq 0$, and so $\sigma(v^\top x) = 0$. Hence,

$$(\sigma(w^\top x) - \sigma(v^\top x)) \mathbb{I}(w^\top x \geq 0) x = \sigma(w^\top x) \mathbb{I}(w^\top x \geq 0) x = \sigma(w^\top x) x$$

where the last equality follows by the fact that $\mathbb{I}(w^\top x \geq 0)$ is redundant given that $\sigma(w^\top x) = 0$ if $w^\top x < 0$. Therefore, for almost every $x \in \text{Uniform}(-L, L) \times \{1\}$, we have that

$$(\sigma(w^\top x) - \sigma(v^\top x)) \mathbb{I}(w^\top x \geq 0) x = \sigma(w^\top x) x \tag{10}$$

We recall that

$$\nabla F(w) = \mathbb{E}_x[(\sigma(w^\top x) - \sigma(v^\top x))\mathbb{I}(w^\top x \geq 0)x]$$

and hence by (10)

$$\nabla F(w) = \mathbb{E}_x[\sigma(w^\top x)x]$$

□

**Lemma E.3.** Suppose that $x$ is sampled according to $x \sim \text{Uniform}(-L, L) \times \{1\} \subseteq \mathbb{R}^2$ and that the target parameters $v = (\tilde{v}, b_v)$ satisfy $\tilde{v} > 0$ and $b_v \leq 0$. then $\nabla F(\cdot)$ is $(L^2 + 1)$-Lipschitz continuous on $w \leq 0$. That is, for any $w, u \leq 0$, $\|\nabla F(w) - \nabla F(u)\| \leq (L^2 + 1)\|w - u\|$.

*Proof.* By $w, u \leq 0$ and Lemma E.2 we have that $\nabla F(w) = \mathbb{E}_x[\sigma(w^\top x)x]$ and $\nabla F(u) = \mathbb{E}_x[\sigma(u^\top x)x]$. Then we establish the desired result as follows,

$$
\begin{aligned}
\|\nabla F(w) - \nabla F(u)\| &= \|\mathbb{E}_x[\sigma(w^\top x)x - \sigma(u^\top x)x]\| \\
&\leq \mathbb{E}_x[\|\sigma(w^\top x)x - \sigma(u^\top x)x\|] && \text{by Jensen's inequality} \\
&= \mathbb{E}_x[|\sigma(w^\top x) - \sigma(u^\top x)| \cdot \|x\|] \\
&\leq \mathbb{E}_x[|w^\top x - u^\top x| \cdot \|x\|] && \text{by Lemma E.1} \\
&\leq \|w - u\|\mathbb{E}_x[\|x\|^2] && \text{by Cauchy-Schwarz inequality} \\
&\leq (L^2 + 1)\|w - u\| && \text{by } x \in [-L, L] \times \{1\}
\end{aligned}
$$

□

### E.2.2 CONVERGENCE AFTER A NEGATIVE INITIALIZATION

**Lemma E.4.** Suppose that $x$ is sampled according to $x \sim \text{Uniform}(-L, L) \times \{1\} \subseteq \mathbb{R}^2$ and that the target parameters $v = (\tilde{v}, b_v)$ satisfy $\tilde{v} > 0$ and $b_v \leq 0$. Then applying gradient descent with L2 Init regularization with strength $\lambda \geq 0$ according to equation (6) with $w_0$ satisfying $\tilde{w}_0 \in [-1, 0)$ and $b_{w_0} = 0$ and learning rate $\eta = \frac{\alpha}{L^2+1}$ where $\alpha \in (0, 1]$ such that $\eta\lambda < 1$, then for any $t \geq 0$

$$\tilde{w}_0 \leq \tilde{w}_t \leq 0, \tag{11}$$
$$b_{w_t} \leq 0, \tag{12}$$
$$F(w_t) \geq F(0) \tag{13}$$

*Proof.* We additionally prove the following invariant

$$\forall t \geq 0, w_t^\top(-L, 1) \geq 0 \tag{14}$$

For $t = 0$ we have that $\tilde{w}_0 \leq \tilde{w}_0 \leq 0$ and $b_{w_0} \leq 0$ hold trivially by assumption. As for (14), we observe that

$$
\begin{aligned}
w_0^\top(-L, 1) &= -L\tilde{w}_0 + b_{w_0} \\
&= -L\tilde{w}_0 && \text{since } b_{w_0} = 0 \\
&\geq 0 && \text{since } \tilde{w}_0 \in [-L, 0)
\end{aligned}
$$

Therefore, we suppose that (11), (12), and (14) hold for some arbitrary $t$ and proceed to show that they hold for $t + 1$. We begin with establishing, (12).

$$
\begin{aligned}
b_{w_{t+1}} &= b_{w_t} - \eta\nabla\bar{F}(w_t)_2 \\
&= b_{w_t} - \eta\nabla F(w_t)_2 - \eta\lambda(b_{w_t} - b_{w_0}) \\
&= (1 - \eta\lambda)b_{w_t} - \eta\nabla F(w_t)_2 && \text{by } b_{w_0} = 0 \\
&= (1 - \eta\lambda)b_{w_t} - \eta\mathbb{E}_x[\sigma(w_t^\top x)] && \text{by Lemma E.2 given that } w_t \leq 0, \tilde{v} > 0, b_v \leq 0 \\
&\leq (1 - \eta\lambda)b_{w_t} && \text{since } \sigma(w_t^\top x) \geq 0, \forall x \\
&\leq 0 && \text{since } b_{w_t} \leq 0 \text{ and } \eta\lambda < 1
\end{aligned}
$$

Next, for (11) we have that $\tilde{w}_{t+1} \leq 0$ if $-L\tilde{w}_{t+1} + b_{w_{t+1}} = w_{t+1}^\top(-L, 1) \geq 0$ since $b_{w_{t+1}} \leq 0$. This is equivalent to showing that

$$(w_t - \eta\nabla F(w_t) - \eta\lambda(w_t - w_0))^\top(-L, 1) \geq 0 \qquad (15)$$

We proceed by bounding $\eta\nabla F(w_t)^\top(-L, 1)$. Again invoking Lemma E.2, we have that

$$
\begin{aligned}
\eta\nabla F(w_t)^\top(-L, 1) &= \eta\mathbb{E}_x[\sigma(w_t^\top x)x]^\top(-L, 1) \\
&= \eta\mathbb{E}_x[\sigma(w_t^\top x)(-L\tilde{x} + 1)] \\
&\leq \eta\sigma(w_t^\top(-L, 1))(L^2 + 1) && \text{by assumption that } \tilde{w}_t \leq 0 \\
&= \alpha\sigma(w_t^\top(-L, 1)) && \text{by choice of } \eta \\
&= \alpha w_t^\top(-L, 1) && \text{by assumption of the invariant (14)} \\
&\leq w_t^\top(-L, 1) && \text{since } \alpha \leq 1
\end{aligned}
$$

Then, we have that

$$
\begin{aligned}
(w_t - \eta\nabla F(w_t) - \eta\lambda(w_t - w_0))^\top(-L, 1) &\geq (1 - \eta\lambda)w_t^\top(-L, 1) + \eta\lambda w_0^\top(-L, 1) - w_t^\top(-L, 1) \\
&\geq (1 - \eta\lambda)w_t^\top(-L, 1) + \eta\lambda w_t^\top(-L, 1) - w_t^\top(-L, 1) \\
&= 0
\end{aligned}
$$

where the second inequality above follows by $\tilde{w}_0 \leq \tilde{w}_t \leq 0$ and $b_{w_t} \leq 0 = b_{w_0}$. Therefore, it follows that $\tilde{w}_{t+1} \leq 0$. Moreover, this also establishes (14) for $t + 1$. As for showing that $\tilde{w}_{t+1} \geq \tilde{w}_0$, in order to complete (11), we argue as follows.

$$
\begin{aligned}
\tilde{w}_{t+1} &= \tilde{w}_t - \eta\nabla\bar{F}(w_t)_1 \\
&= \tilde{w}_t - \eta\nabla F(w_t)_1 - \eta\lambda(\tilde{w}_t - \tilde{w}_0) \\
&= (1 - \eta\lambda)\tilde{w}_t + \eta\lambda\tilde{w}_0 - \eta\mathbb{E}_x[\sigma(w_t^\top x)\tilde{x}] && \text{by Lemma E.2 given that } w_t \leq 0, \tilde{v} > 0, b_v \leq 0 \\
&\geq (1 - \eta\lambda)\tilde{w}_t + \eta\lambda\tilde{w}_0 && \text{since } \tilde{x} > 0 \Rightarrow \sigma(w_t^\top x) = 0 \text{ by } w_t \leq 0 \\
&\geq (1 - \eta\lambda)\tilde{w}_0 + \eta\lambda\tilde{w}_0 && \text{by } \tilde{w}_t \geq \tilde{w}_0 \text{ and } \eta\lambda < 1 \\
&= \tilde{w}_0
\end{aligned}
$$

Therefore, (11) holds for $t + 1$. As for (13), given that we have established (11) and (12) it follows that for any $t \geq 0$

$$
\begin{aligned}
F(w_t) &= \mathbb{E}_x[\frac{1}{2}(\sigma(w_t^\top x) - \sigma(v^\top x))^2] \\
&\geq \mathbb{E}_x[\frac{1}{2}(\sigma(w_t^\top x) - \sigma(v^\top x))^2\mathbb{I}_{\{\tilde{x}\geq 0\}}] \\
&= \mathbb{E}_x[\frac{1}{2}(\sigma(v^\top x))^2\mathbb{I}_{\{\tilde{x}\geq 0\}}] && \text{since } w_t \leq 0 \text{ and hence } \sigma(w_t^\top x)\mathbb{I}_{\{\tilde{x}\geq 0\}} = 0 \\
&= \mathbb{E}_x[\frac{1}{2}(\sigma(v^\top x))^2] && \text{since } v_t^\top x \leq 0 \text{ for } \tilde{x} < 0 \text{ by } \tilde{v} > 0, b_v \leq 0 \\
&= F(0)
\end{aligned}
$$

$\square$

### E.2.3 RESETS AFTER A NEGATIVE INITIALIZATION

**Lemma E.5.** Suppose that $x$ is sampled according to $x \sim \text{Uniform}(-L, L) \times \{1\} \subseteq \mathbb{R}^2$ and that the target parameters $v = (\tilde{v}, b_v)$ satisfy $\tilde{v} > 0$ and $b_v \leq 0$. Then applying gradient descent according to equation (5) with $w_0$ satisfying $\tilde{w}_0 \in [-1, 0]$ and $b_{w_0} = 0$ and learning rate $\eta < \frac{1}{L^2+1}$, we have that for any threshold $\epsilon > 0$, there exists some $t \in \{0, 1, \ldots, T = \lceil\frac{16L^2C_\eta}{\epsilon^4}\rceil\}$, where $C_\eta = \frac{2}{\eta}F((-1, 0))$, such that $\sup_{x\in[-L,L]\times\{1\}} \sigma(w_t^\top x) \leq \epsilon$.

*Proof.* If $\tilde{w}_0 \in [-\frac{\epsilon}{L}, 0]$ then

$$\sup_{x\in[-L,L]\times\{1\}} \sigma(w_t^\top x) = -L\tilde{w}_0 \leq \epsilon$$

and so at $t = 0$ we immediately have that $\sup_{x \in [-L, L] \times \{1\}} \sigma(w_t^\top x) \leq \epsilon$. Therefore, we consider the case of $\tilde{w}_0 \in [-L, -\frac{\epsilon}{L})$ and proceed as follows. According to Lemma E.4 and by the choice of learning rate $\eta$, we have that $\forall t \in \{0, 1, \ldots, T\}, w_t \leq 0$. Therefore by Lemma E.2, $\nabla F(\cdot)$ is $(L^2 + 1)$-Lipschitz continuous on the span of $\{w_0, w_1, \ldots, w_T\}$. Consequently, following the canonical analysis of gradient descent, we have that for any $t \geq 0$

$$F(w_{t+1}) \leq F(w_t) + \nabla F(w_t)^\top (w_{t+1} - w_t) + \frac{L^2 + 1}{2} ||w_{t+1} - w_t||^2$$

$$= F(w_t) - (\eta - \frac{(L^2 + 1)\eta^2}{2})||\nabla F(w_t)||^2$$

$$\leq F(w_t) - \frac{\eta}{2}||\nabla F(w_t)||^2$$

where the second line above follows by $w_{t+1} = w_t - \eta \nabla F(w_t)$ and the last line above follows by the fact that

$$(\eta - \frac{L^2 + 1}{2}\eta^2) - \frac{\eta}{2} = \frac{1}{2}(\eta - (L^2 + 1)\eta^2)$$

$$\geq \frac{1}{2}(\eta - \eta) \qquad\qquad \text{by } \eta \leq \frac{1}{L^2 + 1}$$

$$= 0$$

Then by a telescoping sum,

$$\sum_{t=0}^{\top} ||\nabla F(w_t)||^2 \leq \frac{2}{\eta}(F(w_0) - F(w_{T+1}))$$

$$\leq \frac{2}{\eta} F((-1, 0))$$

where the second line follows by the fact that the loss $F(w_0)$ is maximized at $w_0 = (-1, 0)$ (over the space of initializations of $w_0$) and by the fact that the loss $F(w_{T+1})$ is nonnegative. Then defining $C_\eta = \frac{2}{\eta} F((-1, 0))$, we have that for some $t \in \{0, 1, \ldots, T\}, ||\nabla F(w_t)||^2 \leq \frac{C_\eta}{T}$. More precisely, we have that

$$||\mathbb{E}_x[(\sigma(w^\top x) - \sigma(v^\top x))\mathbb{I}(w^\top x \geq 0)x]||^2 \leq \frac{C_\eta}{T}$$

Given that $x \sim \text{Uniform}(-L, L) \times \{1\}$, by considering the second element of the gradient, which corresponds to the constant component of $x$, this implies that

$$\mathbb{E}_x[(\sigma(w_t^\top x) - \sigma(v^\top x))\mathbb{I}(w^\top x \geq 0)]^2 \leq \frac{C_\eta}{T}$$

Additionally, by Lemma E.4 we have that $\tilde{w}_0 \leq \tilde{w}_t \leq 0$ and $b_{w_t} \leq 0$, and thus by Lemma E.2, we have that

$$\mathbb{E}_x[\sigma(w^\top x)]^2 = \mathbb{E}_x[(\sigma(w^\top x) - \sigma(v^\top x))\mathbb{I}(w_t^\top x \geq 0)]^2 \leq \frac{C_\eta}{T}$$

Hence,

$$\mathbb{E}_x[\sigma(w_t^\top x)] \leq \sqrt{\frac{C_\eta}{T}}$$

Then, noting that $\tilde{w}_0 \leq \tilde{w}_t \leq 0$ and $b_{w_t} \leq 0$, we have that $2L\mathbb{E}_x[\sigma(w_t^\top x)]$ is the area of the triangle formed by the line $w_t^\top x$ over the $x$-axis with its base ranging from its $x$-intercept to $-L$. We denote the length of its base by $b$ and its height by $h$. Given that $\tilde{w}_t \leq 0$ it follows that $h = \sigma(w_t^\top(-L, 1)) = \sup_{x \in [-L, L] \times \{1\}} \sigma(w_t^\top x)$. Additionally, we have that $h = |\tilde{w}_t|b \leq |\tilde{w}_0|b \leq b$ given that $-1 \leq \tilde{w}_0 \leq \tilde{w}_t \leq 0$. Hence we have that

$$\frac{h^2}{4L} \leq \frac{bh}{4L} = \mathbb{E}_x[\sigma(w_t^\top x)] \leq \sqrt{\frac{C_\eta}{T}}$$

By the choice of $T$ we have that $4L\sqrt{\frac{C_\eta}{T}} \leq \epsilon^2$. Hence, we obtain

$$\sup_{x \in [-L, L] \times \{1\}} \sigma(w_t^\top x) = h \leq \sqrt{4L\sqrt{\frac{C_\eta}{T}}} \leq \epsilon$$

$\square$

### E.2.4 CONVERGENCE AFTER A POSITIVE INITIALIZATION

**Lemma E.6.** Suppose that $x$ is sampled according to $x \sim \text{Uniform}(-L, L) \times \{1\} \subseteq \mathbb{R}^2$, the target parameters $v = (\tilde{v}, b_v)$ satisfy $\tilde{v} \in [1, \tilde{v}_{\max}]$ and $-\frac{L}{6} \leq b_v \leq 0$, and initial parameters $w_0$ satisfy $\tilde{w}_0 \in (0, 1]$ and $b_{w_0} = 0$. Then there exists some $\delta \geq \frac{\tilde{w}_0 L^2}{24}$ such that $F(w_0) \leq F(0) - \delta$.

*Proof.*

$$F(w_0) = \mathbb{E}_x[\frac{1}{2}(\sigma(w_0^\top x) - \sigma(v^\top x))^2]$$

$$= F(0) + \frac{1}{2}\mathbb{E}_x[\sigma(w_0^\top x)^2] - \mathbb{E}_x[\sigma(w_0^\top x)\sigma(v^\top x)]$$

$$= F(0) + \frac{\tilde{w}_0^2}{4L}\int_0^L x^2 dx - \mathbb{E}_x[\sigma(w_0^\top x)\sigma(v^\top x)] \qquad \text{by } \tilde{w}_0 \geq 0 \text{ and } b_{w_0} = 0$$

$$= F(0) + \frac{\tilde{w}_0^2 L^2}{12} - \mathbb{E}_x[\sigma(w_0^\top x)\sigma(v^\top x)]$$

Therefore, we define $\delta = -\frac{\tilde{w}_0^2 L^2}{12} + \mathbb{E}_x[\sigma(w_0^\top x)\sigma(v^\top x)]$ and we proceed to show that $\delta \geq \frac{\tilde{w}_0 L^2}{24}$. We let $z = -\frac{b_v}{\tilde{v}}$ so that $(z, 1)$ is the $x$-intercept of the line $v^\top x$. Then, $v^\top x = \tilde{v}(\tilde{x} - z)$. Moreover, $z \in [0, \frac{L}{6}]$ since $b_v \in [-\frac{L}{6}, 0]$ and $\tilde{v} \geq 1$. Therefore, $\forall \tilde{x} \geq z, v^\top x \geq 0$ and $\forall \tilde{x} < z, v^\top x < 0$. Since $b_{w_0} = 0$ and $\tilde{w}_0 \geq 0$, then $\forall \tilde{x} \geq 0, w_0^\top x \geq 0$ and $\forall \tilde{x} < 0, w_0^\top x < 0$. Thus,

$$\sigma(w_0^\top x)\sigma(v^\top x) = \begin{cases} \tilde{w}_0 \tilde{x} \tilde{v}(\tilde{x} - z) & \text{if } \tilde{x} \geq z \\ 0 & \text{if } \tilde{x} < z \end{cases} \tag{16}$$

From here, we can bound the second term of $\delta$ as follows,

$$\mathbb{E}_x[\sigma(w_0^\top x)\sigma(v^\top x)] = \frac{1}{2L}\tilde{w}_0\tilde{v}\int_z^L y^2 - yz dy \qquad \text{by (16)}$$

$$= \frac{1}{2L}\tilde{w}_0\tilde{v}(\frac{L^3}{3} - \frac{L^2 z}{2} + \frac{z^3}{6})$$

Then,

$$\delta = -\frac{\tilde{w}_0^2 L^2}{12} + \mathbb{E}_x[\sigma(w_0^\top x)\sigma(v^\top x)]$$

$$= -\frac{\tilde{w}_0^2 L^2}{12} + \frac{1}{2L}\tilde{w}_0\tilde{v}(\frac{L^3}{3} - \frac{L^2 z}{2} + \frac{z^3}{6})$$

$$\geq -\frac{\tilde{w}_0 L^2}{12} + \frac{1}{2L}\tilde{w}_0\tilde{v}(\frac{L^3}{3} - \frac{L^3}{12}) \qquad \text{by } \frac{\tilde{w}_0\tilde{v}}{2L} \geq 0, z \in [0, \frac{L}{6}]$$

$$= -\frac{\tilde{w}_0^2 L^2}{12} + \frac{\tilde{w}_0\tilde{v}_0 L^2}{8}$$

$$= \frac{\tilde{w}_0 L^2}{4}(-\frac{\tilde{w}_0}{3} + \frac{\tilde{v}}{2})$$

$$\geq \frac{\tilde{w}_0 L^2}{24} \qquad \text{by } \tilde{v} \geq 1, \tilde{w}_0 \leq 1$$

Thus,

$$F(w_0) = F(0) - \delta = \leq F(0) - \frac{\tilde{w}_0 L^2}{24}$$

$\square$

**Lemma E.7.** Suppose that $x$ is sampled according to $x \sim \text{Uniform}(-L, L) \times \{1\} \subseteq \mathbb{R}^2$, the target parameters $v = (\tilde{v}, b_v)$ satisfy $\tilde{v} \in [1, \tilde{v}_{\max}]$ and $-\frac{L}{6} \leq b_v \leq 0$, and initial parameters $w_0$ satisfy $\tilde{w}_0 \in (0, 1]$ and $b_{w_0} = 0$. Then there exists some $\delta \geq \frac{\tilde{w}_0 L^2}{24}$ such that $F(w_0) \leq F(0) - \delta$. Then

defining $\gamma = \frac{\delta^3}{3 \cdot 12^2 (\|w_0\| + \|v\| + 1)^5 (L^2 + 1)^4}$ and applying gradient descent according to equation (5) with learning rate $\eta \leq \frac{\gamma}{(L^2+1)^2}$ we have that $\forall t \geq 0$

$$\|w_t - v\|^2 \leq (1 - \eta\gamma)^t \|w_0 - v\|^2 \tag{17}$$

$$F(w_t) \leq \frac{1}{2}(L^2 + 1)(1 - \eta\gamma)^t \|w_0 - v\|^2 \tag{18}$$

*Proof.* By Lemma E.6, there exists some $\delta \geq \frac{\tilde{w}_0 L^2}{24}$ such that $F(w_0) \leq F(0) - \delta$. Then (17) follows by the latter inequality and a slight modification of the proof of Theorem 5.2 (along with Lemma D.2, Lemma D.4, and Lemma D.6) of Vardi et al. (2021), to extend the result to target parameters $v$ with arbitrary magnitudes. Specifically, a simple modification of the aforementioned proofs implies that setting $\gamma = \frac{\delta^3}{3 \cdot 12^2 (\|w_0\| + \|v\| + 1)^5 c^8 c'^2}$ where we take $c = \max_x \|x\| = \sqrt{L^2 + 1}$ and $c' = 1$ due to $x \sim$ Uniform$(-L, L) \times \{1\}$ guarantees (17). Finally, (18) follows by

$$F(w_t) = \mathbb{E}_x[\frac{1}{2}(\sigma(w_t^\top x) - \sigma(v^\top x))^2]$$

$$\leq \mathbb{E}_x[\frac{1}{2}(w_t^\top x - v^\top x)^2] \qquad \text{by } \sigma(\cdot) \text{ 1-Lipschitz continuous, Lemma E.1}$$

$$\leq \frac{1}{2}\|w_t - v\|^2 \mathbb{E}_x[\|x\|^2] \qquad \text{by the Cauchy Schwarz inequality}$$

$$\leq \frac{1}{2}(L^2 + 1)\|w_t - v\|^2 \qquad \text{by } x \in [-L, L] \times \{1\}$$

$$\leq \frac{1}{2}(L^2 + 1)(1 - \eta\lambda)^t \|w_0 - v\|^2 \qquad \text{by (17)}$$

$\square$

### E.2.5 NO RESETS AFTER A POSITIVE INITIALIZATION

**Lemma E.8.** Suppose that $x$ is sampled according to $x \sim$ Uniform$(-L, L) \times \{1\} \subseteq \mathbb{R}^2$, the target parameters $v = (\tilde{v}, b_v)$ satisfy $\tilde{v} \in [1, \tilde{v}_{\max}]$ and $-\frac{L}{6} \leq b_v \leq 0$. Let $\epsilon > 0$ such that $3(L - z) \geq \frac{\epsilon}{\tilde{v}}$, then for any $w \in \mathbb{R}^2$ such that

$$\sup_{x \in [-L, L] \times \{1\}} w^\top x \leq \epsilon \Rightarrow F(w) \geq F(0) - \frac{\tilde{v}L}{2}\epsilon$$

*Proof.*

$$F(w_t) = \mathbb{E}_x[(\sigma(w^\top x) - \sigma(v^\top x))^2]$$

$$\geq \mathbb{E}_x[(\sigma(w^\top x) - \sigma(v^\top x))^2 \mathbb{I}_{\{v^\top x \geq \epsilon\}}]$$

$$\geq \mathbb{E}_x[(\sigma(v^\top x) - \epsilon)^2 \mathbb{I}_{\{v^\top x \geq \epsilon\}}] \qquad \text{by } \sup_{x \in [-L, L] \times \{1\}} w^\top x \leq \epsilon$$

Let $z = -\frac{b_v}{\tilde{v}}$ such that $v^\top(z, 1) = 0$. Then $v^\top(z + \frac{\epsilon}{\tilde{v}}, 1) = \epsilon$ and since $\tilde{v} > 0$ then $\forall \tilde{x} \geq z + \frac{\epsilon}{\tilde{v}}$ we have that $v^\top(x, 1) \geq \epsilon$. Therefore,

$$\mathbb{E}_x[(\sigma(v^\top x) - \epsilon)^2 \mathbb{I}_{\{v^\top x \geq \epsilon\}}] = \mathbb{E}_x[(v^\top x - \epsilon)^2 \mathbb{I}_{\{v^\top x \geq \epsilon\}}]$$

$$= \frac{1}{2L} \int_{z + \frac{\epsilon}{\tilde{v}}}^L (\tilde{v}y + b_v - \epsilon)^2 dy$$

$$= \frac{1}{2L} \int_0^{L - (z + \frac{\epsilon}{\tilde{v}})} (\tilde{v}y)^2 dy \qquad \text{since } \tilde{v} \geq 0$$

$$= \frac{\tilde{v}^2}{6L}(L - z - \frac{\epsilon}{\tilde{v}})^3$$

By a similar argument, we have that

$$F(0) = \mathbb{E}_x[\sigma(v^\top x)^2]$$

$$= \frac{1}{2L} \int_z^L (\tilde{v}y + b_v)^2 dy$$

$$= \frac{1}{2L} \int_0^{L-z} (\tilde{v}y)^2 dy$$

$$= \frac{\tilde{v}^2}{6L}(L - z)^3$$

Then,

$$F(w) - F(0) \geq \frac{\tilde{v}^2}{6L}((L - z - \frac{\epsilon}{\tilde{v}})^3 - (L - z)^3)$$

$$= \frac{\tilde{v}^2}{6L}(-3(L - z)^2\frac{\epsilon}{\tilde{v}} + 3(L - z)\frac{\epsilon^2}{\tilde{v}^2} - \frac{\epsilon^3}{\tilde{v}^3})$$

$$\geq -\frac{\tilde{v}}{2L}(L - z)^2\epsilon \qquad\qquad \text{since } 3(L - z) \geq \frac{\epsilon}{\tilde{v}}$$

$$\geq -\frac{\tilde{v}L}{2}\epsilon$$

Hence, we have that

$$F(w) \geq F(0) - \frac{\tilde{v}L}{2}\epsilon$$

$\square$

**Lemma E.9.** Suppose that $x$ is sampled according to $x \sim \text{Uniform}(-L, L) \times \{1\} \subseteq \mathbb{R}^2$ and the target parameters $v = (\tilde{v}, b_v)$ satisfy $\tilde{v} \in [1, \tilde{v}_{\max}]$ and $-\frac{L}{6} \leq b_v \leq 0$. Let $\epsilon > 0$ be a reset threshold and suppose that the initial parameters $w_0$ satisfy $\tilde{w}_0 \in (\tilde{w}_{\min}, 1]$ and $b_{w_0} = 0$ such that

$$\tilde{w}_{\min} > \frac{12\tilde{v}\epsilon}{L} \tag{19}$$

and

$$3(L - z) \geq \frac{\epsilon}{\tilde{v}}.$$

Then there exists a $\delta \geq \frac{\tilde{w}_0 L^2}{24}$ such that defining $\gamma = \frac{\delta^3}{3 \cdot 12^2(\|w_0\| + \|v\| + 1)^5(L^2 + 1)^4}$ and applying gradient descent with resets, according to (8) and (9), with learning rate $\eta \leq \frac{\gamma}{(L^2+1)^2}$ we have that $\mathcal{O}_\epsilon(u_{t+1}) = \text{False } \forall t \geq 0$.

*Proof.* According to Lemma E.7, there exists some $\delta \geq \frac{\tilde{w}_0 L^2}{24}$ such that

$$F(w_0) \leq F(0) - \delta$$

Additionally, Lemma E.7, which utilizes a modified proof of Theorem 5.2 of Vardi et al. (2021), ensures that

$$F(u_{t+1}) \leq F(0) - \delta, \forall t \geq 0$$

For the sake of contradiction, we suppose that for some $t \geq 0$, $\mathcal{O}_\epsilon(u_{t+1}) = \text{True}$, or equivalently, that

$$\sup_{x \in [-L,L] \times \{1\}} u_{t+1}^\top x \leq \epsilon$$

Then according to Lemma E.8,

$$F(u_{t+1}) \geq F(0) - \frac{\tilde{v}L\epsilon}{2}$$

However, we note that

$$\frac{\tilde{w}_0 L^2}{24} \geq \frac{\tilde{w}_{\min} L^2}{24}$$

$$> \frac{\tilde{v}L}{2}\epsilon \qquad\qquad \text{by assumption (19)}$$

Then we have that

$$-\frac{\tilde{w}_0 L^2}{24} \geq F(u_{t+1}) - F(0)$$
$$\geq -\frac{\tilde{v} L}{2} \epsilon$$
$$> -\frac{\tilde{w}_0 L^2}{24}$$

which is a contradiction. Therefore, it must follow that $\forall t \geq 0, \mathcal{O}_\epsilon(u_{t+1}) = \text{False}$. $\square$

### E.2.6 PROOF OF THEOREM E.2

**Theorem E.3** (Restatement of Theorem E.2). Suppose that $x$ is sampled according to $x \sim$ Uniform$(-L, L) \times \{1\} \subseteq \mathbb{R}^2$ and the target parameters $v = (\tilde{v}, b_v)$ satisfy $\tilde{v} \in [1, \tilde{v}_{\max}]$ and $-\frac{L}{6} \leq b_v \leq 0$, where we denote $v_{\max} = (\tilde{v}_{\max}, -\frac{L}{6})$. Let $\epsilon > 0$ be a reset threshold and suppose that the initial parameters $w_0$ are sampled uniformly from $\tilde{w}_0 \sim [-1, -\tilde{w}_{\min}) \cup (\tilde{w}_{\min}, 1]$ and $b_{w_0} = 0$ such that

$$\tilde{w}_{\min} > \frac{12 \tilde{v} \epsilon}{L}$$

and

$$3(L - z) \geq \frac{\epsilon}{\tilde{v}}.$$

Then gradient descent with a constant learning rate of

$$\eta \leq \frac{\tilde{w}_{\min}^3 L^6}{3 \cdot 12^2 \cdot 24^3 (2 + ||v_{\max}||)^5 (L^2 + 1)^6}$$

and with resets, i.e. (8) and (9), attains an average regret of

$$\frac{1}{T} \mathbb{E}[R_T] \leq \frac{C}{T}$$

where

$$C = \lceil \frac{32 L^2 F((-1, 0))}{\eta \epsilon^4} \rceil F((-1, 0)) + \frac{3}{2\eta^2} \left( (\tilde{v}_{\max} - \tilde{w}_{\min})^2 \right) + (\frac{L}{6})^2 \right)$$

*Proof.* In order to apply Lemmas E.5, E.7, and E.9 we verify that

$$\eta \leq \frac{1}{L^2 + 1} \tag{20}$$

and

$$\eta \leq \frac{\gamma}{(L^2 + 1)^2} \tag{21}$$

where $\gamma = \frac{\delta^3}{3 \cdot 12^2 (||w_0|| + ||v|| + 1)^5 (L^2 + 1)^4}$ and $\delta \geq \frac{\tilde{w}_0 L^2}{24}$ and where $w_0$ is an arbitrary (re)initialization of $w$ given the prior distribution. For (20) we note that $\tilde{w}_{\min} \leq 1 \leq 3 \cdot 12^2 \cdot 24^3, (2 + ||v_{\max}||)$ and so

$$\eta = \frac{\tilde{w}_{\min}^3 L^6}{3 \cdot 12^2 \cdot 24^3 (2 + ||v_{\max}||)^5 (L^2 + 1)^6}$$
$$\leq \frac{L^6}{(L^2 + 1)^6}$$
$$\leq \frac{(L^2 + 1)^3}{(L^2 + 1)^6}$$
$$= \frac{1}{(L^2 + 1)^3}$$
$$\leq \frac{1}{L^2 + 1}$$

As for (21) we proceed as follows

$$
\begin{aligned}
\eta &= \frac{\tilde{w}_{\min}^3 L^6}{3 \cdot 12^2 \cdot 24^3 (2 + ||v_{\max}||)^5 (L^2 + 1)^6} \\
&\leq \frac{\tilde{w}_0^3 L^6}{3 \cdot 12^2 \cdot 24^3 (||w_0|| + ||v|| + 1)^5 (L^2 + 1)^6} \\
&\leq \frac{\delta^3}{3 \cdot 12^2 (||w_0|| + ||v|| + 1)^5 (L^2 + 1)^6} \qquad \text{by } \delta \geq \frac{\tilde{w}_0 L^2}{24} \\
&= \frac{\gamma}{(L^2 + 1)^2}
\end{aligned}
$$

We continue by upper bounding the regret as follows

$$
\mathbb{E}[R_T] \leq \sum_{r=0}^{\infty} \mathbb{E}[R_T | r \text{ resets occur}]
$$

By Lemma E.5 and Lemma E.9 and the choice of learning rate $\eta$, we only ever reset if $\tilde{w}_t$ is (re)initialized such that $\tilde{w}_t < 0$. Conversely, if we (re)initialize the parameters such that $\tilde{w}_t > 0$ then we never reset for subsequent gradient descent updates. Additionally, since $w_t$ is reinitialized by an independent draw from its initial distribution, then for any particular reset count $r$ we have that

$$
\mathbb{P}(r \text{ resets occur}) = \mathbb{P}(\tilde{w}_0 < 0)^r \cdot \mathbb{P}(\tilde{w}_0 > 0) = \frac{1}{2^{r+1}}
$$

Where $\mathbb{P}(\tilde{w}_0 < 0) = \mathbb{P}(\tilde{w}_0 > 0) = \frac{1}{2}$ by the fact $\tilde{w}_0$ is sampled uniformly from $[-1, -\tilde{w}_{\min}) \cup (\tilde{w}_{\min}, 1]$. Hence, we have that

$$
\mathbb{E}[R_T | r \text{ resets occur}] \leq \mathbb{P}(r \text{ resets occur})(rM_- + M_+) = \frac{1}{2^{r+1}}(rM_- + M_+) \qquad (22)
$$

Where $M_-$ is an upper bound on the total loss during any period of consecutive time steps $t$ to $t'$ such that $w_t$ is a (re)initialization of $w$ such that $\tilde{w}_t < 0$ and $t'$ is the earliest reset after $t$. While $M_+$ is an upper bound on the total loss for time periods after and including $t$ where $w_t$ is a (re)initialization of $w$ such that $\tilde{w}_t > 0$.

By the choice of learning rate $\eta$ and Lemma E.5, if $w$ is (re)initialized such that $\tilde{w}_t < 0$ then within $T_{\text{reset}} \leq \lceil \frac{32 L^2 F((-1,0))}{\eta \epsilon^4} \rceil$ gradient descent updates $w$ is reset. Moreover, the loss $F(w_{t'})$ is at most $F((-1,0))$ for time steps $t'$ preceding a reset of $w$ as loss is maximized at $w_t = (-1, 0)$ over initializations and choosing learning rate $\eta$ at most $\frac{1}{L^2+1}$, the Lipschitz constant of $\nabla F$ (Lemma E.3), guarantees that loss never exceeds $F((-1,0))$. Thus,

$$
M_- \leq T_{\text{reset}} F((-1,0)) \qquad (23)
$$

As for $M_+$ we can construct the following upper bound

$$
\begin{aligned}
M_+ &\leq \mathbb{E}_{w_0}\left[ \sum_{t=0}^{\infty} F(w_t) | \tilde{w}_0 > 0 \right] \\
&\leq \sum_{t=0}^{\infty} (L^2 + 1)(1 - \eta\gamma)^t \mathbb{E}_{w_0}[||w_0 - v||^2 | \tilde{w}_0 > 0] \qquad \text{by Lemma E.7} \\
&= \frac{L^2 + 1}{\eta\gamma} \mathbb{E}_{w_0}[||w_0 - v||^2 | \tilde{w}_0 > 0] \qquad \text{by geometric series} \\
&\leq \frac{1}{\eta\gamma}(L^2 + 1)\left( (\tilde{v}_{\max} - \tilde{w}_{\min})^2 + (\frac{L}{6})^2 \right) \qquad \text{by assumption on } w_0, v \\
&\leq \frac{1}{\eta^2}\left( (\tilde{v}_{\max} - \tilde{w}_{\min})^2 + (\frac{L}{6})^2 \right) \qquad \text{by } \eta \leq \frac{\gamma}{L^2 + 1}
\end{aligned}
$$

Where the use of the geometric series in the third line above is valid by

$$
\eta\gamma < 1
$$

by the assumptions on $\eta$ and $\gamma$. Then returning to our goal of bounding the average regret, we observe that

$$
\begin{aligned}
\mathbb{E}[R_T] &\leq \sum_{r=0}^{\infty} \mathbb{E}[R_T | r \text{ resets occur}] \\
&\leq \sum_{r=0}^{\infty} \frac{1}{2^{r+1}} (r M_- + M_+) && \text{by (22)} \\
&= \frac{1}{2} M_+ + (M_- + M_+) \sum_{r=0}^{\infty} \frac{r}{2^{r+1}} \\
&= \frac{1}{2} M_+ + (M_- + M_+) && \text{by arithmetico-geometric series} \\
&= M_- + \frac{3}{2} M_+
\end{aligned}
$$

Then defining $C = M_- + \frac{3}{2} M_+$, we have the desired result of

$$
\frac{1}{T} \mathbb{E}[R_T] \leq \frac{C}{T}
$$

$\square$

