# OpenReview forum: "Self-Normalized Resets for Plasticity in Continual Learning"
_ICLR.cc/2025/Conference — ICLR 2025 Poster_

### Official Review · Reviewer_PBeJ · 2024-11-02

**Soundness:** 2
**Presentation:** 3
**Contribution:** 2
**Rating:** 5
**Confidence:** 3

**Summary:**

The paper presents a reset-based approach to continual learning, where a network's neurons are reset based on a function of their inactivity. A different threshold function is applied to each neuron. The proposed method is compared to other reset schemes as well as regularization-based approaches. Experiments show promising performance on several datasets (such as MNIST-adjacent ones). Some theory is presented in a setting for learning a single target ReLU.

**Strengths:**

The proposed method appears to outperform the selected baselines on a variety of datasets and exhibits good dependence on its sole hyperparameter. The experiments also appear to be rather thorough. For that reason, I think the proposed method may be practical.

The writing is also strong.

**Weaknesses:**

The experiments appear to be in a rather small setting, which is a slight problem since I believe the paper's main contribution is an experimental one.

Section 4 (theoretical analysis) studies an exceedingly simplified setting. There is no notion of distribution shift or changing tasks (as the expectation is taken with respect to \mu rather than a \mu_t), and the algorithm is full gradient descent, which is not only impractical in real-world settings but impossible without knowledge of the underlying distribution. The second (subtracted) term in the regret also appears to be 0. The restrictions on v in Theorem 4.1 seem rather restrictive and I don't know if they are necessary. It's also unclear if the algorithm suggested in Theorem 4.2 is executable - are the variables with 'min' and 'max' known a priori?

No conclusion section.

If the authors provide strong answers to my concerns, I would be happy to raise my score.

**Questions:**

Section 2 - Each (x_t, y_t) is drawn from a separate distribution \mu_t? Is there any relationship between the different \mu_t?
Line 134 - This looks like dynamic regret, why is that (pessimistic) objective considered?
Line 136 -  Is this the definition of plasticity? That the averaged dynamic regret grows faster than a constant?
Line 144 - What is the formal definition of a 'dead' or 'inactive' neuron?
Line 153/5 - The definitions of Z and A are unclear. What exactly is s indexing? Which two consecutive activations for A?
Line 155 - 'let let' typo
Line 194 - what guarantees do you have on this approximation? \mu_t seems to be the distribution at a particular time, for which you only have one sample or a batch of samples? Why is considering 'firing times' while training on different, past, tasks/distributions a principled thing to do?

How exactly is A_i^{\mu_t} approximated? I would like more detail than footnote 2.

---

> ### Author Response · Authors · 2024-11-28
> **Response to PBeJ**
>
> We appreciate the reviewer’s detailed questions with respect to our mathematical definition of plasticity loss and the theory section. This feedback has been beneficial in crystallizing our presentation in these relevant sections.
>
>
> - $\textbf{PBeJ :}$: “The experiments appear to be in a rather small setting, which is a slight problem since I believe the paper's main contribution is an experimental one.”
>
> - $\textbf{Our Response:}$ We ran a new scaled experiment (256x) and point out in Appendix A.1 that our larger experiments are on par with the largest models considered in the nascent literature on this topic.
> - We wholeheartedly agree that even larger experiments would be of interest but perhaps through the lens of understanding the precise scaling behavior of the plasticity loss phenomenon, which remains unclear (and from our experiments not necessarily aligned with generic scaling laws).
> - Parenthetically, the challenge in scaling in this literature is essentially time: in particular, each experiment typically involves a sequence of between 500 and several thousand tasks which must be visited in sequence. In particular, this is akin to training 500 to several thousand models, each with up to 5 million parameters, in sequence, over multiple seeds and hyper-parameter settings. Despite what we believe is a significantly more efficient implementation of these experiments than what was available when we began this research, a single experiment at our largest scale takes weeks.
>
> $\newline$
>
> - $\textbf{PBeJ :}$ “Section 4 (theoretical analysis) studies an exceedingly simplified setting. There is no notion of distribution shift or changing tasks (as the expectation is taken with respect to \mu rather than a \mu_t), and the algorithm is full gradient descent, which is not only impractical in real-world settings but impossible without knowledge of the underlying distribution. The second (subtracted) term in the regret also appears to be 0. The restrictions on v in Theorem 4.1 seem rather restrictive and I don't know if they are necessary. It's also unclear if the algorithm suggested in Theorem 4.2 is executable - are the variables with 'min' and 'max' known a priori?”
> - $\textbf{Our Response:}$ The empirical research on plasticity loss relates this phenomenon to the scenario where the optimal model weights for a particular task serve as poor initial weights for model training on the next task. The theory in Section 4 thus simply cares to show that (1) regularization based methods will not solve this problem —i.e. there exist weight initializations that will prevent these methods from escaping local minima which is an important point in light of literature suggesting that this will suffice and (2) methods that have access to a ‘reset’ oracle will quickly escape these local minima.
> — You are correct that the algorithm analyzed in Section 4 eschews practical details (eg. using full gradients etc.) to make this point. The setup/assumptions on this, knowing bounds on weights, etc. are akin to the setup in [1] which is a state of the art learning theory result for learning a ReLU. As such, while you are correct that this is not intended to be a theory paper, we would respectfully not view the result as trivial either in execution or utility.
> — To complement the oracular nature of the reset method in Section 4, our theory in Section 2 describes, essentially from the lens of change point detection, why what we do is principled/ optimal, at least from the perspective of detecting a change in the firing rate of a neuron. There we show an optimal hypothesis test, and further now prove a new result (Prop 2.2) showing that the error rates under this new approach to resetting can be substantially lower than prior approaches.
>
> $\newline$
>
> - $\textbf{PBeJ :}$ "No conclusion section."
> - $\textbf{Our Response:}$ Added.
>
> $\newline$
>
> - $ \textbf{PBeJ :}$ "Section 2 - Each (x_t, y_t) is drawn from a separate distribution \mu_t? Is there any relationship between the different \mu_t?"
> - $\textbf{Our Response:}$ For the purposes of our experiments, and the extant literature, they are piecewise constant. See edited footnote 1.
> $\newline$
>
> [1] Vardi, Gal, Gilad Yehudai, and Ohad Shamir. "Learning a single neuron with bias using gradient descent." Advances in Neural Information Processing Systems 34 (2021): 28690-28700.

---

> > ### Comment · Reviewer_PBeJ · 2024-12-01
> >
> > I appreciate the new experiments, but a 5 million parameter model is still quite small by modern standards. I am not suggesting large-scale experiments just to understand scaling behavior, but to actually see if the proposed method works at modern scales. I also don't know why the networks are trained for so many epochs - something like 100 does not seem standard for a language setting at least. I also don't understand why "500 and several thousand tasks" are needed. Unless the authors can point me to an important, real-world scenario where this setting is appropriate, I am not convinced that the paper has studied a meaningful experimental regime.
> >
> > I appreciate the pointer to [1], but I am still unsure about the theoretical setting. How does it capture distribution shift (which is critical for this problem)? If it does, this needs to be made clear. Why are the settings of Theorems E.1 and E.2 not the same (unless I'm misunderstanding something)? If they are not the same, why is this necessary? In general, I need to be convinced that the results here are indeed meaningful for the problem of plasticity loss and changing tasks.
> >
> > "we would respectfully not view the result as trivial either in execution" - Is the proposed algorithm executable?
> >
> > I am happy to raise my score if the authors provide strong answers.

---

> ### Author Response · Authors · 2024-11-28
> **Response to PBeJ II**
>
> - $ \textbf{PBeJ :}$ "Line 134 - This looks like dynamic regret, why is that (pessimistic) objective considered?”
> - $\textbf{Our Response:}$ Surprisingly, the extant literature, being more or less entirely empirical, has not made a precise objective clear, but the dynamic regret objective is indeed what is considered implicitly by the plurality of the literature. It is easy to see why by considering the setting where the sequence of measure $\mu_t$ is piecewise constant; see also footnote 1.
>
> $\newline$
>
> - $ \textbf{PBeJ :}$ “Line 136 - Is this the definition of plasticity? That the averaged dynamic regret grows faster than a constant?”
> - $\textbf{Our Response:}$ This is the definition of plasticity loss that has effectively been adopted by the literature.
>
> $\newline$
>
> - $ \textbf{PBeJ :}$  “Line 144 - What is the formal definition of a 'dead' or 'inactive' neuron?”
> - $\textbf{Our Response:}$ We defined it in Section 2.1 (this is again a term that has not been defined precisely in the literature thus far).
>
> $\newline$
>
> - $ \textbf{PBeJ :}$ “Line 153/5 - The definitions of Z and A are unclear. What exactly is s indexing? Which two consecutive activations for A?”
> - $\textbf{Our Response:}$ We have clarified the definition of these variables, see the paragraph at line 152.
>
> $\newline$
>
>
> - $ \textbf{PBeJ :}$ “Line 155 - 'let let' typo”
> - $\textbf{Our Response:}$ Fixed.
>
> $\newline$
>
> - $ \textbf{PBeJ :}$ “Line 194 - what guarantees do you have on this approximation? \mu_t seems to be the distribution at a particular time, for which you only have one sample or a batch of samples? Why is considering 'firing times' while training on different, past, tasks/distributions a principled thing to do? How exactly is A_i^{\mu_t} approximated? I would like more detail than footnote 2.”
> - $\textbf{Our Response:}$ We have not established formal guarantees on our approximation, but anticipate that the approach in [2] could be used to accomplish that. This would require adopting a full blown change-point detection setup which is cumbersome.
>
>
> [2] Besbes, Omar, and Assaf Zeevi. "On the minimax complexity of pricing in a changing environment." Operations research 59.1 (2011): 66-79.

---

> > ### Author Response · Authors · 2024-12-01
> > **Gentle Reminder**
> >
> > Dear Reviewer,
> >
> > We would like to thank you for the time spent reviewing our submission.
> >
> > The discussion phase will be ending tomorrow, and we wanted to send this gentle reminder. We have done our best to answer the comments you raised, as well as incorporate your suggestions. We would love to hear back from the reviewer and whether we have addressed their concerns.

---

> ### Author Response · Authors · 2024-12-02
>
> "I am still unsure about the theoretical setting. How does it capture distribution shift (which is critical for this problem)?"
>
> Our theoretical analysis, Theorems E.1 and E.2, can be viewed as analyzing the behavior of gradient descent with regularization and reset techniques under a two step training process. We can view the sampling of $w_0$ as a cold-start initialization of a single-ReLU network, or more aptly for a continual learning setting, as **the parameters learned in a preceding task**. We can then view $v$ as the target ReLU that an adversary selects when moving from the first task to the second task.
>
> Theorem E.1 states that an adversary could select a particular target ReLU for the second task, namely any $v$ such that $b_v < 0$ and $\tilde{v} > 0$, which would ensure that with probability at least 0.5, over the parameters $w_0$ learned in the previous task, L2 regularization and L2 Init fail to converge to the optimal parameters $v$ and thus attain constant average regret.
>
> On the other hand Theorem E.2 states that regardless which target ReLU the adversary selects for the second task, provided it is bounded (in magnitude and away from zero), gradient descent with access to a reset oracle attains $O(1/T)$ regret.
>
> These theorems provide a simple example demonstrating that plasticity loss, in a two-task setting, can manifest by the fact that weights learned in a preceding task, $w_0$, can be poor initializations in a secondary task converging to poor local minima which commonly deployed regularization techniques fail to escape. While access to a reset oracle provides a mechanism that effectively learns the target ReLU with a regret guarantee of $O(1/T)$.
>
>
> "Why are the settings of Theorems E.1 and E.2 not the same (unless I'm misunderstanding something)? If they are not the same, why is this necessary?"
>
> The assumptions in E.1 are more general than those in E.2 and so the assumptions in E.2 imply those in E.1.  We state Theorem E.1 with simpler and more general assumptions as the more specific assumptions of E.2 are not necessary for establishing Theorem E.1 and their inclusion would confuse the reader.
>
> The assumptions on Theorem E.2 are more restrictive due to the fact that ensuring an $O(1/T)$ deterministic rate of convergence, of $w_t$ to the target $v$ requires that the target ReLU parameters $v$ have sufficient regularity, namely that $v$ is not arbitrarily small, bounded away from zero, and not arbitrarily large. Note, we could have stated that $\tilde{v}$ must lie in [-\tilde{v}_{max},-1] \cup [1,\tilde{v}_{max}] making the target ReLU symmetric around the origin. Similarly, the theorem requires that the initialization of parameters $w_0$ is bounded and not arbitrarily close to zero. We could relax this condition on sampling $w_0$ away from zero, but to avoid a ``with high probability'' guarantee for simplicity of exposition, we explicitly limit our sampling of $w_0$ to a distribution that is bounded away from zero.
>
> For instance, results in [1] and [3] are all with high probability results that guarantee convergence of learning a single ReLU. We could mimic this flavour of exposition and our assumptions would change to "$w_0$ is sampled uniformly from $[-B,B]$" or ``$w_0$ is sampled from a Gaussian distribution $N(0,\sigma^2)$'' resulting in a regret guarantee with probability at least $1-\Delta$ for some appropriate and diminishing $\Delta$.
>
>  In terms of concerns of practicality, theoretical guarantees are generally proven with respect to ``nice distributions'' such as sampling features from N(0,I) Gaussians as in Theorem 3.1 in [3]. Moreover, restricting the target ReLU parameters to some bounded set ensures, for instance, at minimum that the loss function's gradient enjoys a bounded Lipschitz constant. Tying to practical or real settings, data is almost never sampled from a Guassian distribution nor is the distribution known, and nonetheless results like [1] and [3] are established for Gaussian features and are meaningful contributions. Additionally, constraining the target ReLU to a bounded range (including away from zero) is not unreasonable when features are often standardized and labels are limited to reasonable values, such as 0-1 for classification or for regression problems which could be bounded by natural constants known a-priori with domain-specific knowledge.
>
> [3] Soltanolkotabi, Mahdi. "Learning relus via gradient descent." Advances in neural information processing systems 30 (2017).

---

> > ### Comment · Reviewer_PBeJ · 2024-12-02
> >
> > Thank you for your answers.
> >
> > Theory:
> >
> > 1. I see some negative results in [1] - are you suggesting that your algorithm is better than standard gradient descent for learning a single ReLU?
> > 2. Can the issues presented with the L2 method be solved by picking a random initialization for the 'second task'? How does your analysis account for any sort of transfer that one might expect and hope for in a continual learning setup? Why is it 'necessary' to start from the parameters learned from the previous task?
> >
> > Experiments:
> >
> > I acknowledge that previous work in this area may have studied a similar experimental regime. However, if one is training for so many epochs, what is the benefit of starting from old model parameters? Can't these plasticity issues be resolved by picking a random initialization and training for the same (large) number of epochs? I am concerned about the measures taken to 'induce' this problem of plasticity and if they are natural.

---

> ### Author Response · Authors · 2024-12-02
>
> We would also like to add that, to our knowledge, the theory presented in our paper is the first attempt to model plasticity loss completely by a theoretical or analytical model. Up to this point, there have only been papers that have sought to understand the "causes of plasticity loss" purely through empirical means with varying results. For instance, "Understanding plasticity in neural networks" [4] published just last year attempts to elucidate the causes of plasticity through a series of experiments with mixed results. Our analysis provides a mathematical description of plasticity loss at the granularity of a single neuron when transitioning from one task to another.  Zooming out to the level of a whole neural network, this provides intuition as to why plasticity loss, namely increasing average expected regret, and neuron death occur as a neural network is faced with additional tasks, or distribution shifts.
>
> "Is the proposed algorithm executable?"
>
>  We assume that the reviewer is referring here to Theorem E.2. We should take the results of this section as explaining phenomena in the rubric of plasticity loss "after the fact" in the same manner that much of learning theory aims to understand phenomena in deep learning. For instance, our use of a ``reset oracle'' or supposing a bounded target ReLU are akin to supposing smooth activations as in [5] where instead in practice non-smooth activations such as the ReLU are common.
>
> The reset oracle is to be viewed as an assumption that simplifies our analysis in order to more easily gain an understanding of the phenomenon being modeled. In practice, one approximates this reset oracle through an algorithm such as, our contribution, SNR or by existing methods such as ReDO and CBP.
>
> As for requiring the knowledge of v_{min} and v_{max} these assumptions are to ensure sufficient regularity of the target ReLU and in turn the loss function, as mentioned earlier. Typical convergence proofs for (stochastic) gradient descent suppose, for instance, an L-Lipschitz continuous gradient and in turn demand a learning rate that is at most O(1/L). In practice one never knows the Lipschitz constant of the gradient of the objective function, as this is an intractable computation of the network and training data. Our assumptions in Theorem E.2 are analogous to this fact whereby v_{min} and v_{max} are parameters that govern the regularity of the loss function and are necessary to establish theoretical guarantees but, like the Lipschitz constant of the gradient of the loss of a neural network trained on a particular dataset, are perhaps unknown to the practitioner. Nonetheless, Theorem E.2 states that if the learning rate is chosen to be sufficiently small, akin to choosing an O(1/L) learning rate in a (S)GD analysis, then with resets gradient descent attains O(1/T) average expected regret after a distribution shift.
>
> [4] Lyle, Clare, et al. "Understanding plasticity in neural networks." International Conference on Machine Learning. PMLR, 2023.
>
> [5] Jacot, Arthur, Franck Gabriel, and Clément Hongler. "Neural tangent kernel: Convergence and generalization in neural networks." Advances in neural information processing systems 31 (2018).

---

> ### Author Response · Authors · 2024-12-02
>
> "5 million parameter model is still quite small by modern standards. I am not suggesting large-scale experiments just to understand scaling behavior, but to actually see if the proposed method works at modern scales. I also don't know why the networks are trained for so many epochs - something like 100 does not seem standard for a language setting at least. I also don't understand why "500 and several thousand tasks" are needed."
>
>
> At all model scales or capacities, plasticity loss manifests provided that a network is trained continuously on a sufficient amount of data, see for instance [4] and [8] which have investigated plasticity loss and model capacity. As a consequence, the community has generally found it sufficient to study and address plasticity loss at scales similar to those in our paper in order to run exhaustive ablations with the purpose of carefully understanding the phenomenon and the effect of different interventions rather than simply solving a particular continual learning problem at scale.
>
> In the literature, plasticity loss is often induced by over-training or saturating a network on each individual task out of a sequence of tasks arriving in sequential order. This manifests in experiments that train a network over hundreds of tasks, where on each task the network is trained for hundreds of epochs. This is an experimental framework for plasticity loss that has been well studied in the literature and we are not the first to propose such a training regime. For instance, a recent paper published in Nature this year [6] trains a ResNet-18 architecture over the CIFAR-100 dataset for 200 epochs per task over a sequence of 20 tasks. This paper also considers the Permuted MNIST and Continual ImageNet problems which consist of thousands of consecutive tasks and demonstrate that training over such a large sequence of tasks can be necessary to producing plasticity loss, in comparison to earlier papers which demonstrated plasticity loss, of a different flavour, in 2-stage training regimes [15]. See references [4,6-15] which are all papers in recent years that study plasticity loss and train networks for hundreds of epochs on datasets no larger than those considered in our paper.
>
> [4] Lyle, Clare, et al. "Understanding plasticity in neural networks." International Conference on Machine Learning. PMLR, 2023.
>
> [6] Dohare, Shibhansh, et al. "Loss of plasticity in deep continual learning." Nature 632.8026 (2024): 768-774.
>
>
> [7] Kumar, Saurabh, Henrik Marklund, and Benjamin Van Roy. "Maintaining plasticity via regenerative regularization." arXiv preprint arXiv:2308.11958 (2023).
>
>
>
> [8] Kumar, Saurabh, et al. "Continual learning as computationally constrained reinforcement learning." arXiv preprint arXiv:2307.04345 (2023).
>
>
>
> [9] Lee, Hojoon, et al. "Slow and Steady Wins the Race: Maintaining Plasticity with Hare and Tortoise Networks." Forty-first International Conference on Machine Learning.
>
>
>
> [10] Elsayed, Mohamed, et al. "Weight Clipping for Deep Continual and Reinforcement Learning." Reinforcement Learning Conference.
>
> [11] Lyle, Clare, et al. "Disentangling the causes of plasticity loss in neural networks." arXiv preprint arXiv:2402.18762 (2024).
>
> [12] Nikishin, E., Schwarzer, M., D’Oro, P., Bacon, P. L., & Courville, A. (2022, June). The primacy bias in deep reinforcement learning. In International conference on machine learning (pp. 16828-16847). PMLR.
>
> [13] Nikishin, E., Oh, J., Ostrovski, G., Lyle, C., Pascanu, R., Dabney, W., & Barreto, A. (2024). Deep reinforcement learning with plasticity injection. Advances in Neural Information Processing Systems, 36.
>
>
> [14] Lewandowski, A., Tanaka, H., Schuurmans, D., & Machado, M. C. (2023). Curvature Explains Loss of Plasticity. arXiv preprint arXiv:2312.00246.
>
> [15] Ash, Jordan, and Ryan P. Adams. "On warm-starting neural network training." Advances in neural information processing systems 33 (2020): 3884-3894.

---

> ### Author Response · Authors · 2024-12-02
>
> Thank you for the response. These are good questions that get to the heart of continual learning.
>
> "I see some negative results in [1] - are you suggesting that your algorithm is better than standard gradient descent for learning a single ReLU?"
>
> **Response:** Our Theorem E.1 is analogous to the negative results in [1], except proven for our restricted dimension and additionally extended for L2 regularization and L2 Init. Yes, our algorithm is better for learning a single ReLU compared to gradient descent as gradient descent can get "stuck" according to the negative result in [1], or equivalently, our Theorem E.1. Whereas, our algorithm, analyzed with an idealized oracle in Theorem E.2, can learn a ReLU over all initializations since anytime it get's "stuck" in a bad local minimum it will reset.
>
> "Can the issues presented with the L2 method be solved by picking a random initialization for the 'second task'?"
>
> **Response:** No, resetting with L2 **only once** for the second task will still result in picking a bad initialization with probability 0.5 (in our analytical model). To completely eliminate this issue with L2 one would need to deploy a "reset oracle" that can reinitialize the weights whenever (S)GD enters a bad region -- this amounts to combining our algorithm with L2 regularization.
>
> "How does your analysis account for any sort of transfer that one might expect and hope for in a continual learning setup? Why is it 'necessary' to start from the parameters learned from the previous task?"
>
> **Response:** There is some nuance here when discussing plasticity loss and transfer learning. Plasticity loss is a phenomenon that emerges during warm-start training, transfer learning, or continual learning, all of which are names for situations where a network is trained on a dataset or set of tasks and is then trained once more on a second dataset of set of tasks, or potentially repeating this process multiple times. It is 'necessary' to start from the parameters learned from the previous task because in applications where tasks are related, such as fine tuning a language model or vision model, parameters learned on earlier tasks provide "good" representations for the subsequent or down stream task, which may often have few data points to train on.
>
> When investigating plasticity loss, empirically, researchers tend to formulate a sequence of tasks, with varying degrees of similarity or differences (e.g. Permuted MNIST vs Continual ImageNet). The metric that is generally optimized is the online train/test accuracy/loss, without explicitly accounting for any 'transfer'. The fact the network is trained on a sequence of tasks makes implicit that transfer is occurring.
>
> Our analysis simply brings this empirical investigation to an analytical model and down to the level of a single neuron being trained from one task to another. Specifically, our analysis considers the case where one samples, or learns on a previous task, a $w_0$ and is then asked to use this $w_0$ to learn a target ReLU, parameterized by $v$, on a subsequent task. As we want to model plasticity loss through the perspective that solutions for one task can be poor initializations for a subsequent task, as has been demonstrated empirically in [4,6-15], then $v$ is selected adversarially. Just like the preceding empirical investigations in [4,6-15], our analysis doesn't necessarily "care" about transfer as the objective is to minimize (some definition of) regret. The only place where "transfer" can appear in this analysis is in the fact that $w_0$ is some set of weights that are selected for the subsequent task due to the expectation that task 1 and task 2 are related and hence weights in one task should be "good" for the second task, although not necessarily always so.
>
> In fact, it is not clear how to model 'transfer' when learning a single neuron from one task to another, besides our setup of supposing that $w_0$ is some weight vector learned on the preceding task(s). Moreover, as argued above, 'transfer' is implicit when studying plasticity loss and is not some objective or metric that is being measured or optimized. Primarily, plasticity loss is the phenomenon of increasing regret when repeatedly performing transfer learning, so implicit transfer is occurring, and the issue to resolve is "how do we perform transfer learning (or continual learning)" while maintaining plasticity, i.e. minimizing regret on each subsequent task. In other words "how do we use the weights from a previous task to minimize regret on the current task?"
>
> Our analysis finds that sometimes these weights from task 1 can be poor initializations for task 2 and an effective remedy is to simply reinitialize the weights once it is deemed that GD is stuck in a bad local minimum.

---

> ### Author Response · Authors · 2024-12-02
>
> **Perhaps, if one really wanted to explicitly model the necessity of transfer and plasticity loss, one could define $v$ as follows.**
> - **With probability 0.5, $v = w_0$.**
> -  **With probability 0.5, $v$ is selected adversarially.**
>
> The latter case where $v = w_0$ is relatively trivial to analyze and makes clear that transfer is beneficial. The former case where $v$ is selected adversarially is precisely the story that we have laid out and demonstrates that (L2/L2 Init regularized) gradient descent can get stuck in a "poor" local minimum, i.e. experience plasticity loss, while an algorithm with access to a reset oracle attains $O(1/T)$ regret.

---

> > ### Author Response · Authors · 2024-12-02
> >
> > "I acknowledge that previous work in this area may have studied a similar experimental regime. However, if one is training for so many epochs, what is the benefit of starting from old model parameters? Can't these plasticity issues be resolved by picking a random initialization and training for the same (large) number of epochs? I am concerned about the measures taken to 'induce' this problem of plasticity and if they are natural."
> >
> > **Response:**  I do agree that training for many epochs could be somewhat 'unnatural', however, there are several practical reasons, from a research perspective, as to why many experimental regimes consist of training for hundreds of epochs.
> >
> > - **Highlighting the Phenomenon:** Training extensively on one task can lead to overfitting to that task, which in turn can exacerbate plasticity loss. This makes the loss of the network's ability to learn new tasks more pronounced and easier to observe and analyze.
> >
> > - **Simulating Real-world Scenarios:** In practical applications, models may encounter scenarios where they are exposed to certain tasks or data distributions for extended periods. Training for many epochs mimics these conditions, providing insights into how models might perform over time.
> >
> > - **Remove Confounding of Undertraining:** To remove the confounding effect of undertraining, networks are trained on each task to convergence. Overtraining a bit ensures that the network cannot improve further and assigning a generous and fixed epoch count for each task ensures sufficient convergence for each task and a comparable resource allocation. This ensures that any increasing regret over time is due to plasticity loss and not due to any other confounders.
> >
> > - **Extreme But Necessary Cases:** At minimum, any algorithm that mitigates plasticity loss should work successfully on a range of datasets, distribution shifts, and architectures in these simpler and more controlled environments. Similarly, at minimum any analysis that seeks to explain plasticity loss should explain the phenomenon in these simpler and more controlled environments that have been investigated in the literature [4,6-15]. **However, the literature is simply not there yet** -- that is, there is no prior continual learning intervention that simply maintains plasticity loss across datasets, distribution shifts, and architectures. See for instance Table 1 in our paper which shows that, besides our contribution SNR, over a set of representative existing benchmark continual learning problems, no single existing continual learning method is robust across problems, even at this "small" scale. In addition, Appendix C demonstrates that all existing methods, except SNR, are sensitive to their choice of hyper parameters, indicating existing methods tuned on one environment or one period of data may fare poorly in a new environment or a new period of data, even for the "simple" or "small" settings considered in the literature. Moreover, the recent work on this problem [4,6-15] indicates that the community has not yet consolidated on a "solution" or "set of solutions" to plasticity loss, even at these "small" scales.
> >
> > We, therefore, would argue that expending resources to deploy a continual learning algorithm in a larger "real-world" setting could result in noisey or confounding results as the performance of the evaluated algorithms may be confounded by idiosyncrasies of the particular models and datasets evaluated. Instead, rigorously resolving the issue of plasticity loss at scales, and perhaps "extreme" but **crucially consistent and replicable training regimes**, seen thus far in the literature gives us a scale that allows for thorough and exhaustive ablations that are necessary for truly elucidating the causes of plasticity loss and measuring the relative merits of different approaches for addressing this phenomenon.

---

> > > ### Comment · Reviewer_PBeJ · 2024-12-02
> > >
> > > I am satisfied with the answers to the theory. I will raise my score if the authors
> > >
> > > 1. Theory: thoroughly integrate the theoretical points made in the rebuttal to the paper, as they are necessary to motivate the theory section and understand its contributions.
> > > 2. Experiments: I understand there is little time left, but I would like to see what happens when the networks are re-initialized randomly before each new task *in the exact same training regime with many epochs*. I would need to see that your method outperforms this baseline. If the current baselines do not outperform it either, then this might suggest the chosen experimental regime is not appropriate for continual learning and plasticity loss.

---

> > > > ### Author Response · Authors · 2024-12-03
> > > >
> > > > 1. Theory: We have actually rephrased the presentation of the theory in our paper, see the blue text in Section 4. But of course, we are happy to incorporate any additional details and clarification from this discussion into the final paper submission.
> > > >
> > > > 2. Experiments: This is a great idea, particularly relevant to the Continual ImageNet (CI) experiment. We have been able to implemented this "cold start" algorithm for this experiment with the Adam optimizer. However, it turns out that other than No Intervention and Layer Norm, all other algorithms outperform cold start. We make the point that the implementation of cold start resets is **unfair and impractical** as in practice one is never aware of task boundaries. All other algorithms are agnostic to task boundaries and are therefore realistically implementable. Below are the results for Continual ImageNet with cold start resets:
> > > >
> > > > No Intervention: 0.581 (0.081)
> > > >
> > > > SNR: 0.847 (0.005)
> > > >
> > > > CBP: 0.818 (0.005)
> > > >
> > > > ReDO: 0.803 (0.063)
> > > >
> > > > L2 Reg.: 0.803 (0.009)
> > > >
> > > > L2 Init: 0.827 (0.008)
> > > >
> > > > S&P: 0.814 (0.005)
> > > >
> > > > Layer Norm: 0.651 (0.053)
> > > >
> > > > Cold Start Resets: 0.801 (0.011)
> > > >
> > > > In the full paper we are happy to put in more of these where it makes sense.

---

> > > > > ### Comment · Reviewer_PBeJ · 2024-12-03
> > > > >
> > > > > Thank you, I have raised my score.

---

### Official Review · Reviewer_hG3B · 2024-11-04

**Soundness:** 2
**Presentation:** 2
**Contribution:** 2
**Rating:** 6
**Confidence:** 4

**Summary:**

The authors introduce Self-Normalized Resets (SNR) which is an adaptive algorithm that resets neuron weights when there is a drop in its firing rate in order to reduce plasticity loss. They also conduct a theoretical investigation of the optimization landscape for learning a single ReLU neuron. Their empirical results show that while existing baselines struggle in terms of sensitivity and detecting inactive neurons, SNR is robust to hyperparameter tuning, and can result in better performance.

**Strengths:**

- As resetting neurons is a well-known technique in addressing plasticity loss, the main novelty of this paper lies in its exploration of the question of 'how to reset' the neurons. If I understood correctly, by focusing on the *recent* activity of a neuron rather than activity within a uniform period of time, SNR tries to answer the question by selecting individual neurons to reset by being more sensitive to the distribution shifts.
- Their empirical results (especially Table 1) clearly show better average accuracy on a variety of benchmarks. In addition, SNR is better in terms of robustness to its hyper-parameter as compared to other baselines.

**Weaknesses:**

- While I understand the adaptive nature of SNR in selecting neurons, wouldn't it also result in over-dependence on the type of distribution shifts across tasks? The vision experiments, in particular, are primarily conducted on benchmarks where a task is uniformly sampled. As a result, the boundaries (or shifts) between consecutive tasks are relatively uniform as well. But what if instead of uniformly, the tasks arrive based on increasing/decreasing (or both) levels of noise and difficulty [1]? A controlled experiment could be conducted to analyze how the nominal firing rate of that neuron changes based on such distribution shifts.
- The authors described their method SNR as an alternative way to pick neurons to reset them. However, in their experiments, they did not compare SNR with Redo and CBP in Table 2. Also, the standard deviation is crucially missing in Table 1.
- The paper appears to be incomplete i.e., it ends abruptly after section 4 (with theorem 4.2) without providing any discussion points, limitations, and conclusion.


There are several grammatical/clarity issues -
- Line 215 - tacking the histogram
- Lines 57 - 60 can be explained better.
- Line 269 - Shouldn't it be 1000 classes?
- Line 377 - seems redundant after the previous para

**Questions:**

- How would SNR's test accuracy (on the held-out dataset) compare to other reset methods? Recent works have been focusing on generalization performance showing that while reset-based methods may result in better training accuracy, they often struggle in terms of generalizability and are outperformed by regularization methods like L2 [1].
- Why do some of the methods in Table 1 results show a significant drop in performance compared to others? Is it dependent on the size/type of hyper-parameter search?

**I am willing to increase the score if the authors address the concerns and questions raised.**

[1] Hojoon Lee, Hyeonseo Cho, Hyunseung Kim, Donghu Kim, Dugki Min, Jaegul Choo, and Clare Lyle. Slow and steady wins the race: Maintaining plasticity with hare and tortoise networks. arXiv preprint arXiv:2406.02596, 2024.

---

> ### Author Response · Authors · 2024-11-28
> **Response to Reviewer hG3B**
>
> We appreciate the reviewer’s thoughful review. We have added several new experiments and additional discussion and analysis in response to the feedback.
>
> - $\textbf{hG3B:}$ “While I understand the adaptive nature of SNR in selecting neurons, wouldn't it also result in over-dependence on the type of distribution shifts across tasks? […] But what if instead of uniformly, the tasks arrive based on increasing/decreasing (or both) levels of noise and difficulty [1]?”
>
>  - $\textbf{Our Response:}$ We conducted a new suite of experiments with decreasing label noise in the spirit on [1]. Thank you for the super interesting pointer. We show (a) similar relative merits to what we had earlier, when measuring training accuracy for SNR and its competitors, and, (b) highlight that when measuring test loss, we must deconfound the impact of epochs per task (or else we will conflate generic overfitting with plasticity loss). We hope you like these! Please see 3.2.2 Generalization on line 461 and A.2 Generalization Results on line 668.
>
> $\newline$
>
> - $\textbf{hG3B:}$  “ The authors described their method SNR as an alternative way to pick neurons to reset them. However, in their experiments, they did not compare SNR with Redo and CBP in Table 2. Also, the standard deviation is crucially missing in Table 1.”
>
>  - $\textbf{Our Response:}$ This was more or less because those methods were not competitive. We have added ReDO and CBP to Table 2. We have also added standard deviations to the results in Table 1, but for brevity, the standard deviations are presented in a larger complete table, Table 6, on line 733 in the Appendix.
>
> $\newline$
>
> - $\textbf{hG3B:}$ “The paper appears to be incomplete i.e., it ends abruptly after section 4 (with theorem 4.2) without providing any discussion points, limitations, and conclusion."
>
>  - $\textbf{Our Response:}$We have added Section 5 Discussion and Limitations to address this point.
>
> $\newline$
>
> - $\textbf{hG3B:}$ "How would SNR's test accuracy (on the held-out dataset) compare to other reset methods? Recent works have been focusing on generalization performance showing that while reset-based methods may result in better training accuracy, they often struggle in terms of generalizability and are outperformed by regularization methods like L2 [1]."
>
>  - $\textbf{Our Response:}$ See our response above; we have included a new set of experiments focused on test accuracy in 3.2.2 Generalization on line 461 and A.2 Generalization Results on line 668.
>
> $\newline$
>
> - $\textbf{hG3B:}$ "Why do some of the methods in Table 1 results show a significant drop in performance compared to others? Is it dependent on the size/type of hyper-parameter search?"
>
>  - $\textbf{Our Response:}$ In fairness to the competing methods, they all improve upon no intervention, and so play some role in alleviating plasticity loss. Our sense for why this is happening with reset methods is simply that they can have varying effectiveness at accurately resetting neurons. The (new) theory in Sec 2.1 tries to shed light on why this is the case.

---

> > ### Comment · Reviewer_hG3B · 2024-12-01
> >
> > Thank you for responding and running the experiments, the authors have addressed most of my concerns. Based on the results in Table 5, it appears that SNR performs slightly better only in the PM-G2. Its performance in the other settings, however, is not particularly remarkable (as also seen in Fig 6). Still, the paper looks in a much better shape now as compared to the initial submission and I will increase my score accordingly.

---

### Official Review · Reviewer_GyG5 · 2024-11-05

**Soundness:** 4
**Presentation:** 3
**Contribution:** 4
**Rating:** 8
**Confidence:** 4

**Summary:**

This work presents a new method for avoiding plasticity loss in continual learning settings (where neural networks lose their ability to learn new tasks after being trained on other tasks). The proposed method is called Self-Normalized Resets (SNR) and simply involves choosing a threshold time of zero activations, so that any neuron that does not activate (has 0 output) for the threshold number of training iterations has its weights randomly reset during training. SNR is found to outperform other methods in a number of small continual learning image classification tasks, and combines with L2 regularization to outperform on a small natural language task.

**Strengths:**

__Originality:__ The idea of resetting neurons to avoid loss of plasticity is not particularly novel or surprising (as evidenced by the cited related work), but the proposed method appears novel and most importantly, satisfies multiple desiderata (simple, effective, robust) well beyond previous methods.

__Quality:__ Although on relatively small tasks, the experiments are thorough and convincing. The comparisons are made on a variety of datasets and consideration is made for hyperparameter tuning. A number of known covariates for plasticity loss (L2 complexity of parameters, dead neurons) are plotted to explain the effectiveness of SNR versus other methods.

__Clarity:__ The paper is overall well written and easy to understand (see Weaknesses for a few areas that need improvement).

__Significance:__ The results are only on small datasets, but nevertheless show promise (scaling Permuted Shakespeare gives a positive trend in terms of the method's superiority). The robustness of SNR to hyperparameter choice is very striking. To highlight this, I actually would recommend putting at least 1 figure or table in the main text to summarize one of the tables in Appendix C: for example, plotting hyperparameter strength vs accuracy on one CL task for the compared methods.

**Weaknesses:**

This paragraph is confusing: "This reveals an interesting comparison with SNR. The schemes above will re-initialize a neuron after inactivity over a period of time that is uniform across all neurons. In the context of the hypothesis testing setup above, this will result in sub-optimal error rates across neurons. On the other hand, SNR will *reset a neuron after it is inactive for a period that is effectively normalized to the nominal firing rate of that neuron*, while still only specifying a single hyperparameter for the network." - referring to the italicized part of the last sentence, doesn't SNR also reset at a uniform rate of $\bar{T}$? The only difference I can tell is that in the previously described method, neurons are reset all at once every $r$ minibatches, and it is possible for a neuron to be inactive for anywhere between $r$ and $2r-1$ minibatches before reset. If $\bar{T} = r$ then SNR only differs in resetting neurons at exactly $r$ minibatches, and resetting individual neurons at different times during training. Could the authors please clarify this point?

In Permuted Shakespeare, what is the actual task being optimized (e.g. classification, next word prediction, etc.)?

Notes on the presentation:
- a discussion/conclusion section is missing
- some figures have too much noise to be easily readable (e.g. figure 3) and could either have a lower sampling rate for the displayed points, or use a confidence interval to cover the range of variability.
- typo on "hyperparemter(s)." (line 331)
- typo in appendix B "with the [second] element fixed as 1" (line 723)
- typo in appendix B: $w_0$ should be $w_{min}$ in the second part of "We sample $w_0$ uniformly from $[−1, w_0 ) \cup (w_0, 1]$" (line 755)

**Questions:**

One issue with the tasks chosen is that they are entirely disjoint and do not test for robustness to distribution shifts or catastrophic forgetting - e.g. training on task A and B alternately and trying to maintain accuracy on A. Although out of the immediate scope of neural plasticity, I think these factors are important to consider for practical settings. How does SNR interact with catastrophic forgetting, i.e. does it make it better or worse?

Is L2+SNR competitive with SNR alone for the tasks other than Permuted Shakespeare? If so, it would be nice to promote L2+SNR as a "default" method that works for all tasks.

---

> ### Author Response · Authors · 2024-11-28
> **Response to GyG5**
>
> We appreciate that the author has found the paper to be clear and substantial. The feedback regarding further clarity with regard to the novelty of SNR's reset mechanism has been beneficial to improving our presentation.
>
> - $\textbf{GyG5 :}$ “This paragraph is confusing: ‘This reveals […] ‘ - […] doesn’t SNR also reset at a uniform rate of T¯? The only difference I can tell is that in the previously described method, neurons are reset all at once every r minibatches, and it is possible for a neuron to be inactive for anywhere between r and 2r−1 minibatches before reset. If T¯=r then SNR only differs in resetting neurons at exactly r minibatches, and resetting individual neurons at different times during training. Could the authors please clarify this point?”
> - $\textbf{Our Response}$ This has been carefully fixed; Section 2.1 has been rewritten to carefully disambiguate the two approaches. We have also added a theoretical result that characterize the error rate under ReDO vs SNR (Prop 2.2).
>
> $\newline$
>
>
> - $\textbf{GyG5 :}$ “In Permuted Shakespeare, what is the actual task being optimized (e.g. classification, next word prediction, etc.)?”
> - $\textbf{Our Response}$ Next word prediction.
>
> $\newline$
>
> - $\textbf{GyG5 :}$ “a discussion/conclusion section is missing”
> - $\textbf{Our Response}$ We have added Section 5 to address this point.
>
> $\newline$
>
> - $\textbf{GyG5 :}$ “One issue with the tasks chosen is that they are entirely disjoint and do not test for robustness to distribution shifts or catastrophic forgetting […] How does SNR interact with catastrophic forgetting, i.e. does it make it better or worse?”
> - $\textbf{Our Response}$ This is a great question! We do not yet have an informed answer.
>
> $\newline$
>
> - $\textbf{GyG5 :}$  “Is L2+SNR competitive with SNR alone for the tasks other than Permuted Shakespeare? If so, it would be nice to promote L2+SNR as a "default" method that works for all tasks.”
>
> - $\textbf{Our Response}$ We suspect L2+SNR would actually be worse. This is now evident from the new ablation (L2*+SNR) where we only regularize the attention mechanism — this outperforms L2+SNR.

---

> > ### Author Response · Authors · 2024-12-01
> > **Gentle Reminder**
> >
> > Dear Reviewer,
> >
> > We would like to thank you for the time spent reviewing our submission.
> >
> > The discussion phase will be ending tomorrow, and we wanted to send this gentle reminder. We have done our best to answer the comments you raised, as well as incorporate your suggestions. We would love to hear back from the reviewer and whether we have addressed their concerns.

---

> > > ### Comment · Reviewer_GyG5 · 2024-12-02
> > >
> > > Thank you for the rebuttals. I have also read the comments by the other reviewers and will stand by my score. Addressing specific comments:
> > >
> > > 1. While the setting is small and artificial, I agree with the authors that a) it is in line with prior work on neural plasticity, and b) it is a sufficient model for the "worst case" that any algorithm addressing neural plasticity must overcome.
> > > 2. While the online or continual learning setting is not experimented on, the method is actually better suited to the online setting than existing reset algorithms because neurons can be reset at any time that they exceed the specified non-firing threshold. This makes the method broadly transferable in application.
> > > 3. The main presentation issues have been addressed in the revision.

---

### Official Review · Reviewer_aGhR · 2024-11-06

**Soundness:** 2
**Presentation:** 1
**Contribution:** 2
**Rating:** 3
**Confidence:** 4

**Summary:**

This paper proposes a plasticity loss mitigation method called Self Normalized Resets (SNR). It proposes a new method to decide when a neuron should be reset. Specifically, for each neuron, it tracks the mean time between non zero activations of that neuron, models a geometric distribution with that mean, and resets the neuron if the probability that a random variable with this distribution is greater than the neuron’s current time without firing is less than some hyperparameter.

The paper performs some experiments on Permuted MNIST, Random Label MNIST, Random Label CIFAR10, Continual Imagenet, and Permuted Shakespeare, showing a slight improvement or matching previous baselines.

**Strengths:**

The paper’s new mechanism to select which neurons to reset seems interesting. The theory and intuition behind the method need to be explained more clearly, but the results do show a slight improvement on some settings for some datasets.

**Weaknesses:**

- This paper feels incomplete. The entire theory section in Section 4 is hard to follow and does not feel very motivated. It’s unclear why Section 4.2 is put in the paper. The paper then immediately ends after an equation, with no explanation of how it relates to the rest of the paper, and it is missing a conclusion section.
- The results in Table 1 are conducted over 5 seeds, but they are missing any error bars.
- Table 2 shows results on a task introduced in this paper, Permuted Shakespeare, but several of the baselines for earlier experiments are missing for this dataset.
- The caption for Figure 3 could be made clearer.
- The details on the hyperparameter search could be clearer.

**Questions:**

- How do you reset the neurons that you select? Is it the same procedure as in ReDo?
- How does this method do at improving generalization/testing accuracy across distribution shifts?

---

> ### Author Response · Authors · 2024-11-28
> **Response to aGhR**
>
> We appreciate the reviewer's careful feedback. In particular, the comments regarding additional clarity and motivation regarding our method have been beneficial to improving the presentation of our paper.
>
> - $\textbf{aGhR :}$ “This paper feels incomplete. The entire theory section in Section 4 is hard to follow […] and it is missing a conclusion section.”
> - $\textbf{Our Response:}$ We have rewritten this. We hope that things are now clear. The net of the theory is to show that regularization is unlikely to suffice to fix plasticity loss issues (Theorem 4.1) whereas certain ‘idealized’ resets help (Theorem 4.1). The theory in Section 2 characterizes SNR as providing an ideal such reset (Prop 2.1). We have also been able to show now that competing reset schemes can incur substantially higher error rates (Prop 2.2).
>
> $\newline$
>
> - $\textbf{aGhR :}$ "The results in Table 1 are conducted over 5 seeds, but they are missing any error bars."
> - $\textbf{Our Response:}$  Fixed.
>
> $\newline$
>
> - $\textbf{aGhR :}$ "Table 2 shows results on a task introduced in this paper, Permuted Shakespeare, but several of the baselines for earlier experiments are missing for this dataset."
> - $\textbf{Our Response:}$  Fixed. We have also added standard deviations to the results in Table 1, but for brevity, the standard deviations are presented in a larger complete table, Table 6, on line 733 in the Appendix.
>
> $\newline$
>
> - $\textbf{aGhR :}$ “The caption for Figure 3 could be made clearer.”
> - $\textbf{Our Response:}$  Fixed.
>
> $\newline$
>
> - $\textbf{aGhR :}$ “The details on the hyperparameter search could be clearer.”
> - $\textbf{Our Response:}$  See Appendix C for the details of our hyperparameter search. In Table 7 in the appendix we list the hyperparameters that were included in our hyper parameter sweep. For each experiment and algorithm we selected the optimal hyperparameter(s) by evaluation on an initial set of random seeds. All results in the main body of the paper are with these optimal hyper parameters on a new set of random seeds.
>
> $\newline$
>
> - $\textbf{aGhR :}$ "How do you reset the neurons that you select? Is it the same procedure as in ReDo?”
> - $\textbf{Our Response:}$  No, it is quite different and a core contrition of the paper. Whereas existing reset schemes relied on specifying ad hoc notions of utility, we cast this problem as one of detecting change points (Section 2.1), and derived an optimal solution (Prop 2.1). That optimal solution is precisely SNR, and we can show theoretically that other other rest schemes (and ReDO) specifically would have substantially higher Type 1 errors on resetting neurons (Prop 2.2). This theory bares out elegantly in our experiments.
>
> $\newline$
>
> - $\textbf{aGhR :}$ "How does this method do at improving generalization/testing accuracy across distribution shifts?"
> - $\textbf{Our Response:}$ Whereas plasticity loss is primarily discussed in the context of training loss, we have added a new section (Secvtion 3.2.2, Appendix A.2) discussing this issue. In short, SNR attains similar merits relative to its competitor algorithms as in the original experiments which using the metric of training accuracy.

---

> > ### Author Response · Authors · 2024-12-01
> > **Gentle Reminder**
> >
> > Dear Reviewer,
> >
> > We would like to thank you for the time spent reviewing our submission.
> >
> > The discussion phase will be ending tomorrow, and we wanted to send this gentle reminder. We have done our best to answer the comments you raised, as well as incorporate your suggestions. We would love to hear back from the reviewer and whether we have addressed their concerns.

---

> > > ### Comment · Reviewer_aGhR · 2024-12-02
> > >
> > > Hi,
> > >
> > > Thank you for the changes made to the paper. I think it did improve the clarity a bit, but there are still a few more points.
> > >
> > > - You mention "The algorithm requires a single hyper parameter, $\eta$." This is not true, however, as you mention that you approximate the distribution of the interfiring time as a Geometric variable with mean calculated based on "a fixed length trailing window $m$" where m is a hyperparameter. Furthermore, how are you able to track the mean of a fixed length window with just a single parameter? Don't you need to track all the numbers in the window?
> > > - Is the approximation about the mean firing time following a geometric distribution validated anywhere, either through your experiments or through prior work?
> > > - Figure 3, both axes say Inactive neurons, but I am guessing one of them is weight norm based on the text.
> > > - One big issue I have is the fact that the main baselines to this method should be ReDo and CBP, and other than mentioning that SNR does better, there isn't much empirical analysis showing the difference between them.
> > > - I might be mistaken, but the argument in Section 2 tries to show that SNR resets neurons that have been firing less frequently than they used to in a previous window. It is not clear to me if this is the "optimal" approach to resetting neurons, although I can appreciate that it might be a better scheme than ReDo.
> > >
> > > Overall, I think this paper needs a bit more development in (1) comparisons to prior work (2) clearer explanations of the theory and why the assumptions made should hold (3) general improvement in writing. I am maintaining my score.

---

> ### Author Response · Authors · 2024-12-03
>
> - "You mention "The algorithm requires a single hyper parameter, ." This is not true, however, as you mention that you approximate the distribution of the interfiring time as a Geometric variable with mean calculated based on "a fixed length trailing window " where m is a hyperparameter."
>
> - **Response:** The window length m **is fixed and is not tuned at all** across all experiments. Therefore SNR is effectively characterized by a single hyperparameter, the rejection percentile threshold \eta. This is not uncommon as across many competitor algorithms there are hyper parameters that are fixed and not tuned, for instance age maturity thresholds and decay factors. Such hyper parameters are often left unaltered from the default values in subsequent papers.
>
> - "Furthermore, how are you able to track the mean of a fixed length window with just a single parameter? Don't you need to track all the numbers in the window?"
>
> - **Response:** We don't fully follow the reference to "a single parameter here". To elaborate, the mean of a geometric random variable can be estimated by tracking two integers: the sum of the arrival times and the number of arrivals. See for instance the section on Statistical Inference in https://en.wikipedia.org/wiki/Geometric_distribution for a simple overview of estimators for the Geometric distribution.
>
> - "Is the approximation about the mean firing time following a geometric distribution validated anywhere, either through your experiments or through prior work?"
>
> - **Response**: A neuron's interfiring time follows a geometric distribution by definition. Specifically, a neuron that fires independently at each time step with probability $p$ must have an inter firing time that is distributed according to a geometric distribution with parameter $p$. In the standard (continual) supervised learning regime, individual examples are always independent, satisfying the assumptions of our model. **See our discussion with reviewer PBeJ.**
>
> - "Figure 3, both axes say Inactive neurons, but I am guessing one of them is weight norm based on the text."
>
> - **Response:** Actually both panels in this plot display the number of inactive neutrons. The right panel is simply the left panel zoomed in so at to distinguish the neuron inactivity between SNR, L2, and SNR-L2. See the first three panels in Figure 4 for weight norms in the Permuted Shakespeare experiment.
>
> - "One big issue I have is the fact that the main baselines to this method should be ReDo and CBP, and other than mentioning that SNR does better, there isn't much empirical analysis showing the difference between them."
>
> - **Response:** Our new Proposition 2.2 states that, even for a simple example with two neurons, using a single reset threshold or frequency, as is the case for ReDO and CBP, the total error rate when reseting neurons can be quite poor. We are making the point here than an optimal reset algorithm, whose purpose is to effectively identify and reset inactive/dormant/dead neutrons, must necessarily reset neutrons rates that are normalized to each individual neuron. At present, existing reset algorithms, namely CBP and ReDO, use fixed reset frequencies (and thresholds) that are incomparable to the self-normalized nature of SNR's reset threshold. We believe that this is a worthwhile conceptualization and analysis of reset based algorithms, as up to this point, any analysis of CBP and ReDO has been entirely empirical with no theoretical analysis.
> We are happy to include plots in the final paper submission showcasing the reset thresholds that SNR "learns" for different neurons and layers over time - demonstrating the necessity of heterogeneity in reset thresholds across neurons and layers.
>
> - "I might be mistaken, but the argument in Section 2 tries to show that SNR resets neurons that have been firing less frequently than they used to in a previous window. It is not clear to me if this is the "optimal" approach to resetting neurons, although I can appreciate that it might be a better scheme than ReDo."
>
> - **Response:** **See our discussion with reviewer PBeJ where we have addressed this point.** In short, in our theoretical analysis of plasticity loss, we have shown that when transitioning from one task to another, the effective firing rate of a neuron can drastically fall, relative to its rate on the previous task. Moreover, this lower firing rate of the neuron occurs in synch with and is the driving force behind plasticity loss, i.e. increasing average regret when training from one task to another. **See the discussion with PBeJ regarding Theorem E.2 and in particular how the reset-oracle is employed to identify the point at which a neuron's firing rate has effectively disappeared, allowing the network to identify poor local minimum, or equivalently, change points from one task to another**.

---

### Official Review · Reviewer_HcgF · 2024-11-09

**Soundness:** 3
**Presentation:** 3
**Contribution:** 2
**Rating:** 5
**Confidence:** 3

**Summary:**

The paper proposes a particular heuristic (SNR) to identify neurons that need to be reset in order to recover plasticity. They show that this particular choice of resetting neurons works very well. The work motivates intuitively how SNR compares to more complex utility measures proposed previously, and show results on a bunch of continual learning problems.

**Strengths:**

The method is simple, easy to implement. And empirically on the problems it has been applied to, it outperforms the existing baseline, which are similar reset methods that use more complex conditions to reset (basically define complex utility functions that are used to make this selection).

**Weaknesses:**

I'm somewhat concerned about the significance of the results. I think the method outperforms the baseline, but most more detailed results tend to be on permuted MNIST or permuted Shakespeare. I know previous work relied on similar small scale problems, and it feels unfair to punish this work, while the others got away with it. But at some point I'm worried that we are reading too much into these numbers.

The authors have run on other tasks (Random Label Cifar, Continual ImageNet) but we can only see an aggregate result on those. And why they highlight specific conditions, they are still borderline as dataset (at least the random label cifar). I think some previous papers also showed results on atari which I think is a more complex setting. And previous works have not scaled things a lot more either.

A second weakness is in my view, not sufficient analysis. E.g. trying to compare some of those utility functions with the SNR heuristic. How often they disagree. Is the intuition that some of these utility functions don't take into account the firing rate of the unit the answer (it the paper there is a suggestion in this direction). The paper argues that L2init is needed for Shakespeare because of the attention layer. Has this been ablated (e.g. apply L2 init just to attention weights?).  Or is the L2init within the MLP block important. And if so why?

Putting them together, I feel that the main sales pitch is that it is a simpler utility function and it got better numbers, but there is less on understanding why this is the right thing to do. And I'm worried at such it might not have the impact on the community it should.

Finally I find the lack of conclusion and discussion a bit strange. I understand that there was a page limit, but I find the solution of not providing a discussion suboptimal. I would urge the authors to find other ways to reduce the paper size.

Overall I'm happy to increase my score if some of the questions are answered.

**Questions:**

1. Could you move other things to the appendix to make room for a conclusion / discussion section? The results section is meant to include the discussion, but I don't think it goes sufficiently deep into the discussion aspect of it.

2. Can you run an ablation showing what happens if you only L2 the attention weights?

3. Can you provide some more insights in how the heuristic works better than the others? Is it that is resets less, therefore is less disruptive. Or the amount of reset correlates better with the firing rate of the unit? Or something else?

4. There has been a bit of discussion in the papers that you cite (e.g. works from Lyle et al.) on the causes of the loss of plasticity. Can you somehow connect a bit more the heuristic and the causes of plasticity? Right now it feels like the heuristic is just more reliable at predicting when a unit is actually dormant rather then firing rarely. But maybe is not as much connected to the problem it tries to resolve, that of loss of plasticity.

---

> ### Author Response · Authors · 2024-11-28
> **Response to HcgF**
>
> We appreciate the reviewer’s thorough and detailed feedback. The comments have been helpful in improving our investigation.
>
> - $\textbf{HcgF :}$ “I’m somewhat concerned about the significance of the results. […] I think some previous papers also showed results on atari which I think is a more complex setting.”
> - $\textbf{Our Response:}$ We agree with the sentiment. This is the reason we have attempted to introduce a new problem (Permuted Shakespeare) which allows for a natural scaling. We have added a further scaling of our largest experiment there (256x or 5.1 million parameters) that is to the best of our knowledge quite a bit larger than the typical problems considered for plasticity loss.
>
> $\newline$
>
>
> - $\textbf{HcgF :}$ “A second weakness is in my view, not sufficient analysis. E.g. trying to compare some of those utility functions with the SNR heuristic.”
> - $\textbf{Our Response:}$ SNR does not rely on utility functions, relying instead on a rigorous hypothesis test. In section 2.1, we now analyze what can go wrong with existing methods showing theoretically that their error rate can be substantially higher than SNR.
>
> $\newline$
>
> - $\textbf{HcgF :}$  “The paper argues that L2init is needed for Shakespeare because of the attention layer. Has this been ablated (e.g. apply L2 init just to attention weights?). Or is the L2init within the MLP block important. And if so why?”
> - $\textbf{Our Response:}$ We assume you mean L2 here, and your idea is a good one. We perform this ablation now, see L2* and SNR+L2* which implement regularization only on attention weights, in Table 2. In fact, it appears that regularization on just the attention layer suffices which actually aligns better with the experiments that precede Permuted Shakespeare and our claim. Parenthetically, we note that as far as we can tell this is the first paper to actually consider the plasticity loss phenomenon carefully in the context of a transformer.
>
> $\newline$
>
> - $\textbf{HcgF :}$ “I feel that the main sales pitch is that it is a simpler utility function and it got better numbers, but there is less on understanding why this is the right thing to do.”
> - $\textbf{Our Response:}$ We have worked to address why we believe the idea here is not a ‘hack’: The empirical research on plasticity loss relates this phenomenon to the scenario where the optimal model weights for a particular task serve as poor initial weights for model training on the next task.
> The theory in Section 4 shows that (1) regularization based methods will not solve this problem —i.e. there exist weight initializations that will prevent these methods from escaping local minima which is an important point in light of literature suggesting that this will suffice and (2) methods that have access to a ‘reset’ oracle will quickly escape these local minima.
> The theory in Section 2 shows that resetting optimally is itself a non-trivial task, and presents a framework showing that SNR (which does not rely on utility functions, but instead a hypothesis test) is in essence an optimal approach to resetting and that the error rates under this approach can be substantially smaller than alternatives.
>
> $\newline$
>
> - $\textbf{HcgF :}$ “Finally I find the lack of conclusion and discussion a bit strange.”
> - $\textbf{Our Response:}$ Fixed
>
> $\newline$
>
> - $\textbf{HcgF :}$ “Could you move other things to the appendix to make room for a conclusion.”
> - $\textbf{Our Response:}$ Done.
>
> $\newline$
>
> - $\textbf{HcgF :}$ “Can you run an ablation showing what happens if you only L2 the attention weights?”
> - $\textbf{Our Response:}$ Done — this was a great idea!Basically, it shows that L2 regularization is only needed on the attention layers (which have no neurons), and for neurons in the MLP layer, resets suffice. A nice clean lesson.
>
> $\newline$
>
>  - $\textbf{HcgF :}$ “Can you provide some more insights in how the heuristic works better than the others?”
> - $\textbf{Our Response:}$ We worked hard to do this. See the new proposition 2.2. Basically, we show that even with two neurons with different nominal firing rates, the error rate under a scheme like ReDO can be substantially larger. This is especially the case in the regime where hyper parameters are tuned to reset infrequently (i.e. with low Type 1 error) and where the neurons have substantially different nominal firing rates.

---

> ### Author Response · Authors · 2024-11-28
> **Response to HcgF II**
>
> - $\textbf{HcgF :}$ “Can you somehow connect a bit more the heuristic and the causes of plasticity?”
> - $\textbf{Our Response:}$ This was actually the goal of Section 4, but the message got lost in our presentation: The empirical research on plasticity loss relates this phenomenon to the scenario where the optimal model weights for a particular task serve as poor initial weights for model training on the next task; i.e. random initializations have high plasticity whereas initializations that correspond to optimal weights for some related task may not.
> - The theory in Section 4 shows that (1) regularization based methods will not solve this problem —i.e. there exist weight initializations that will prevent these methods from escaping local minima which is an important point in light of literature suggesting that this will suffice and (2) methods that have access to a ‘reset’ oracle will quickly escape these local minima.
> - The theory in Section 2 shows that resetting optimally is itself a non-trivial task, and presents a framework showing that SNR (which does not rely on utility functions, but instead a hypothesis test) is in essence an optimal approach to resetting and that the error rates under this approach can be substantially smaller than alternatives.
> - Taken together, these illustrate how SNR addresses a common explanation for plasticity loss (i.e. poor initializations for new tasks).

---

> > ### Author Response · Authors · 2024-12-01
> > **Gentle Reminder**
> >
> > Dear Reviewer,
> >
> > We would like to thank you for the time spent reviewing our submission.
> >
> > The discussion phase will be ending tomorrow, and we wanted to send this gentle reminder. We have done our best to answer the comments you raised, as well as incorporate your suggestions. We would love to hear back from the reviewer and whether we have addressed their concerns.

---

> > > ### Author Response · Authors · 2024-12-03
> > > **Dear Reviewer,**
> > >
> > > The extended discussion phase will be ending soon today, and we wanted to send this gentle reminder. We have done our best to answer the comments you raised, as well as incorporate your suggestions, and those of other reviewers. We would love to hear back from the reviewer and whether we have addressed their concerns.

---

### Author Response · Authors · 2024-11-28
**Revised Paper Submission**

We appreciate the careful feedback from the reviewers. In response to the reviewers' concerns and questions, we have uploaded a revised version of our submission. All changes are highlighted in blue font in order to more easily identify where and how we have addressed the reviewers' comments. We respond to each reviewer in detail below.

---

### Meta-Review · Area_Chair_jDMc · 2024-12-15

**Metareview:**

The paper introduces Self-Normalised Resets (SNR), a novel method to mitigate plasticity loss in continual learning by resetting neurons. The method is simple and requires minimal parameter tuning, yet it outperforms competing methods on standard benchmarks. The authors provide a theoretical argument that L2 regularisation alone is insufficient to prevent plasticity loss, whereas the proposed SNR effectively addresses this issue, offering better performance in maintaining plasticity over time.

**Additional Comments On Reviewer Discussion:**

The reviewers reported 3 main critiques concerning: 1. the experimental validation; 2. the theoretical setting; 3. clarity.
In the discussion period the authors improved their manuscript addressing this critique and/or providing convincing arguments in favour of their setup.

In particular:
1. **Limited experimental validation.** The authors improved the presentation of their experiments and provided additional material in the appendix. Concerning the critique about limited benchmarks, the authors reported several recent works that are using the same benchmarks.
2. **Theoretical setting is oversimplified.** The theory is performend in a simplified setting with a very simple task and simple learning algorithm, as it is also acknowledged by the authors. However they argued showing that the theory provides same meaningful insights and it is not very common in the practical literature.
3. **Claritiy.** The authors improved the presentation of the work in several points of the paper. Limiting the discussion to the main text, they improved the section motivating the method, added details in the theoretical section and included a conclusion section discussing several of the discussion point that emerged during the review process.

---

### Decision · Program_Chairs · 2025-01-22

Accept (Poster)